# Making AI Forget You:
# Data Deletion in Machine Learning

Antonio A. Ginart[1], Melody Y. Guan[2], Gregory Valiant[2], and James Zou[3]

[1]Dept. of Electrical Engineering
[2]Dept. of Computer Science
[3]Dept. of Biomedial Data Science
Stanford University, Palo Alto, CA 94305
{tginart, mguan, valiant, jamesz}@stanford.edu

## Abstract

Intense recent discussions have focused on how to provide individuals with control over when their data can and cannot be used — the EU's Right To Be Forgotten regulation is an example of this effort. In this paper we initiate a framework studying what to do when it is no longer permissible to deploy models derivative from specific user data. In particular, we formulate the problem of efficiently deleting individual data points from trained machine learning models. For many standard ML models, the only way to completely remove an individual's data is to retrain the whole model from scratch on the remaining data, which is often not computationally practical. We investigate algorithmic principles that enable efficient data deletion in ML. For the specific setting of $k$-means clustering, we propose two provably efficient deletion algorithms which achieve an average of over $100\times$ improvement in deletion efficiency across 6 datasets, while producing clusters of comparable statistical quality to a canonical $k$-means++ baseline.

## 1 Introduction

Recently, one of the authors received the redacted email below, informing us that an individual's data cannot be used any longer. The UK Biobank [79] is one of the most valuable collections of genetic and medical records with half a million participants. Thousands of machine learning classifiers are trained on this data, and thousands of papers have been published using this data.

```
EMAIL -- UK BIOBANK --
Subject:  UK Biobank Application [REDACTED], Participant Withdrawal Notification [REDACTED]

Dear Researcher,

As you are aware, participants are free to withdraw form the UK Biobank at any time and request that their
data no longer be used.  Since our last review, some participants involved with Application [REDACTED]
have requested that their data should longer be used.
```

The email request from the UK Biobank illustrates a fundamental challenge the broad data science and policy community is grappling with: *how should we provide individuals with flexible control over how corporations, governments, and researchers use their data?* Individuals could decide at any time that they do not wish for their personal data to be used for a particular purpose by a particular entity. This ability is sometimes legally enforced. For example, the European Union's General Data Protection Regulation (GDPR) and former Right to Be Forgotten [24, 23] both require that companies and organizations enable users to withdraw consent to their data at any time under certain circumstances. These regulations broadly affect international companies and technology platforms with EU customers and users. Legal scholars have pointed out that the continued use of AI systems directly trained on deleted data could be considered illegal under certain interpretations and ultimately concluded that: *it may be impossible to fulfill the legal aims of the Right to be Forgotten in artificial intelligence environments* [86]. Furthermore, so-called *model-inversion attacks* have demonstrated the capability of adversaries to extract user information from trained ML models [85].

Concretely, we frame the problem of data deletion in machine learning as follows. Suppose a statistical model is trained on $n$ datapoints. For example, the model could be trained to perform disease diagnosis from data collected from $n$ patients. To *delete* the data sampled from the $i$-th patient from our trained model, we would like to update it such that it becomes independent of sample $i$, and looks as if it had been trained on the remaining $n-1$ patients. A naive approach to satisfy the requested deletion would be to retrain the model from scratch on the data from the remaining $n-1$ patients. For many applications, this is not a tractable solution – the costs (in time, computation, and energy) for training many machine learning models can be quite high. Large scale algorithms can take weeks to train and consume large amounts of electricity and other resources. Hence, we posit that efficient data deletion is a fundamental data management operation for machine learning models and AI systems, just like in relational databases or other classical data structures.

Beyond supporting individual data rights, there are various other possible use cases in which efficient data deletion is desirable. To name a few examples, it could be used to speed-up leave-one-out-cross-validation [2], support a user data marketplace [75, 80], or identify important or valuable datapoints within a model [37].

Deletion efficiency for general learning algorithms has not been previously studied. While the desired output of a deletion operation on a *deterministic* model is fairly obvious, we have yet to even define data deletion for stochastic learning algorithms. At present, there is only a handful of learning algorithms known to support fast data deletion operations, all of which are deterministic. Even so, there is no pre-existing notion of how engineers should think about the asymptotic *deletion efficiency* of learning systems, nor understanding of the kinds of trade-offs such systems face.

The key components of this paper include introducing deletion efficient learning, based on an intuitive and operational notion of what it means to (efficiently) delete data from a (possibly stochastic) statistical model. We pose data deletion as an online problem, from which a notion of optimal deletion efficiency emerges from a natural lower bound on amortized computation time. We do a case-study on deletion efficient learning using the simple, yet perennial, $k$-means clustering problem. We propose two deletion efficient algorithms that (in certain regimes) achieve optimal deletion efficiency. Empirically, on six datasets, our methods achieve an average of over $100\times$ speedup in amortized runtime with respect to the canonical Lloyd's algorithm seeded by $k$-means++ [53, 5]. Simultaneously, our proposed deletion efficient algorithms perform comparably to the canonical algorithm on three different statistical metrics of clustering quality. Finally, we synthesize an algorithmic toolbox for designing deletion efficient learning systems.

We summarize our work into three contributions:

**(1)** We formalize the problem and notion of efficient data deletion in the context of machine learning.

**(2)** We propose two different deletion efficient solutions for $k$-means clustering that have theoretical guarantees and strong empirical results.

**(3)** From our theory and experiments, we synthesize four general engineering principles for designing deletion efficient learning systems.

## 2  Related Works

**Deterministic Deletion Updates**  As mentioned in the introduction, efficient deletion operations are known for some canonical learning algorithms. They include linear models [55, 27, 83, 81, 18, 74], certain types of *lazy learning* [88, 6, 11] techniques such as non-parametric Nadaraya-Watson kernel regressions [61] or nearest-neighbors methods [22, 74], recursive support vector machines [19, 81], and co-occurrence based collaborative filtering [74].

**Data Deletion and Data Privacy**  Related ideas for protecting data in machine learning — e.g. cryptography [63, 16, 14, 13, 62, 31], and differential privacy [30, 21, 20, 64, 1] — do not lead to efficient data deletion, but rather attempt to make data private or non-identifiable. Algorithms that support efficient deletion do not have to be private, and algorithms that are private do not have to support efficient deletion. To see the difference between privacy and data deletion, note that every learning algorithm supports the naive data deletion operation of retraining from scratch. The algorithm is not required to satisfy any privacy guarantees. *Even an operation that outputs the entire dataset in the clear could support data deletion, whereas such an operation is certainly not private*. In this sense, the challenge of data deletion only arises in the presence of computational limitations. Privacy, on the other hand, presents statistical challenges, even in the absence of any computational limitations. With that being said, data deletion has direct connections and consequences in data privacy and security, which we explore in more detail in Appendix A.

# 3 Problem Formulation

We proceed by describing our setting and defining the notion of *data deletion* in the context of a machine learning algorithm and model. Our definition formalizes the intuitive goal that after a specified datapoint, $x$, is deleted, the resulting model is updated to be indistinguishable from a model that was trained from scratch on the dataset sans $x$. Once we have defined data deletion, we define a notion of *deletion efficiency* in the context of an online setting. Finally, we conclude by synthesizing high-level principles for designing deletion efficient learning algorithms.

Throughout we denote dataset $D = \{x_1,...,x_n\}$ as a set consisting of $n$ datapoints, with each datapoint $x_i \in \mathbf{R}^d$; for simplicity, we often represent $D$ as a $n \times d$ real-valued matrix as well. Let $A$ denote a (possibly randomized) algorithm that maps a dataset to a model in hypothesis space $\mathcal{H}$. We allow models to also include arbitrary metadata that is not necessarily used at inference time. Such metadata could include data structures or partial computations that can be leveraged to help with subsequent deletions. We also emphasize that algorithm $A$ operates on datasets of any size. Since $A$ is often stochastic, we can also treat $A$ as implicitly defining a conditional distribution over $\mathcal{H}$ given dataset $D$.

**Definition 3.1. Data Deletion Operation:** We define a *data deletion* operation for learning algorithm $A$, $R_A(D, A(D), i)$, which maps the dataset $D$, model $A(D)$, and index $i \in \{1,...,n\}$ to some model in $\mathcal{H}$. Such an operation is a data deletion operation if, for all $D$ and $i$, random variables $A(D_{-i})$ and $R_A(D, A(D), i)$ are equal in distribution, $A(D_{-i}) =_d R_A(D, A(D), i)$.

Here we focus on exact data deletion: after deleting a training point from the model, the model should be as if this training point had never been seen in the first place. The above definition can naturally be relaxed to approximate data deletion by requiring a bound on the distance (or divergence) between distributions of $A(D_{-i})$ and $R_A(D, A(D), i)$. Refer to Appendix A for more details on approximate data deletion, especially in connection to differential privacy. We defer a full discussion of this to future work.

**A Computational Challenge**    Every learning algorithm, $A$, supports a trivial data deletion operation corresponding to simply retraining on the new dataset after the specified datapoint has been removed — namely running algorithm $A$ on the dataset $D_{-i}$. Because of this, the challenge of data deletion is computational: **1)** Can we design a learning algorithm $A$, and supporting data structures, so as to allow for a computationally efficient data deletion operation? **2)** For what algorithms $A$ is there a data deletion operation that runs in time sublinear in the size of the dataset, or at least sublinear in the time it takes to compute the original model, $A(D)$? **3)** How do restrictions on the memory-footprint of the metadata contained in $A(D)$ impact the efficiency of data deletion algorithms?

**Data Deletion as an Online Problem**    One convenient way of concretely formulating the computational challenge of data deletion is via the lens of online algorithms [17]. Given a dataset of $n$ datapoints, a specific training algorithm $A$, and its corresponding deletion operation $R_A$, one can consider a stream of $m \leq n$ distinct indices, $i_1, i_2,...,i_m \in \{1,...,n\}$, corresponding to the sequence of datapoints to be deleted. The online task then is to design a data deletion operation that is given the indices $\{i_j\}$ one at a time, and must output $A(D_{-\{i_1,...,i_j\}})$ upon being given index $i_j$. As in the extensive body of work on online algorithms, the goal is to minimize the amortized computation time. The amortized runtime in the proposed online deletion setting is a natural and meaningful way to measure deletion efficiency. A formal definition of our proposed online problem setting can be found in Appendix A.

In online data deletion, a simple lower bound on amortized runtime emerges. All (sequential) learning algorithms $A$ run in time $\Omega(n)$ under the natural assumption that $A$ must process each datapoint at least once. Furthermore, in the best case, $A$ comes with a constant time deletion operation (or a deletion oracle).

**Remark 3.1.** *In the online setting, for $n$ datapoints and $m$ deletion requests we establish an asymptotic lower bound of $\Omega(\frac{n}{m})$ for the amortized computation time of any (sequential) learning algorithm.*

We refer to an algorithm achieving this lower bound as *deletion efficient*. Obtaining tight upper and lower bounds is an open question for many basic learning paradigms including ridge regression, decision tree models, and settings where $A$ corresponds to the solution to a stochastic optimization problem. In this paper, we do a case study on $k$-means clustering, showing that we can achieve deletion efficiency without sacrificing statistical performance.

## 3.1 General Principles for Deletion Efficient Machine Learning Systems

We identify four design principles which we envision as the pillars of deletion efficient learning algorithms.

**Linearity**   Use of linear computation allows for simple post-processing to undo the influence of a single datapoint on a set of parameters. Generally speaking, the Sherman-Morrison-Woodbury matrix identity and matrix factorization techniques can be used to derive fast and explicit formulas for updating linear models [55, 27, 83, 43]. For example, in the case of linear least squares regressions, QR factorization can be used to delete datapoints from learned weights in time $O(d^2)$ [41, 90]. Linearity should be most effective in domains in which randomized [70], reservoir [89, 76], domain-specific [54], or pre-trained feature spaces elucidate linear relationships in the data.

**Laziness**   Lazy learning methods delay computation until inference time [88, 11, 6], resulting in trivial deletions. One of the simplest examples of lazy learning is $k$-nearest neighbors [32, 4, 74], where deleting a point from the dataset at deletion time directly translates to an updated model at inference time. There is a natural affinity between lazy learning and non-parametric techniques [61, 15]. Although we did not make use of laziness for unsupervised learning in this work, pre-existing literature on kernel density estimation for clustering would be a natural starting place [44]. Laziness should be most effective in regimes when there are fewer constraints on inference time and model memory than training time or deletion time. In some sense, laziness can be interpreted as shifting computation from training to inference. As a side effect, deletion can be immensely simplified.

**Modularity**   In the context of deletion efficient learning, modularity is the restriction of dependence of computation state or model parameters to specific partitions of the dataset. Under such a modularization, we can isolate specific modules of data processing that need to be recomputed in order to account for deletions to the dataset. Our notion of modularity is conceptually similar to its use in software design [10] and distributed computing [67]. In DC-$k$-means, we leverage modularity by managing the dependence between computation and data via the divide-and-conquer tree. Modularity should be most effective in regimes for which the dimension of the data is small compared to the dataset size, allowing for partitions of the dataset to capture the important structure and features.

**Quantization**   Many models come with a sense of continuity from dataset space to model space — small changes to the dataset should result in small changes to the (distribution over the) model. In statistical and computational learning theory, this idea is known to as *stability* [60, 47, 50, 29, 77, 68]. We can leverage stability by quantizing the mapping from datasets to models (either explicitly or implicitly). Then, for a small number of deletions, such a quantized model is unlikely to change. If this can be efficiently verified at deletion time, then it can be used for fast average-case deletions. Quantization is most effective in regimes for which the number of parameters is small compared to the dataset size.

## 4   Deletion Efficient Clustering

Data deletion is a general challenge for machine learning. Due to its simplicity we focus on $k$-means clustering as a case study. Clustering is a widely used ML application, including on the UK Biobank (for example as in [33]). We propose two algorithms for deletion efficient $k$-means clustering. In the context of $k$-means, we treat the output centroids as the model from which we are interested in deleting datapoints. We summarize our proposed algorithms and state theoretical runtime complexity and statistical performance guarantees. Please refer to [32] for background concerning $k$-means clustering.

### 4.1   Quantized $k$-Means

We propose a quantized variant of Lloyd's algorithm as a deletion efficient solution to $k$-means clustering, called Q-$k$-means. By quantizing the centroids at each iteration, we show that the algorithm's centroids are constant with respect to deletions with high probability. Under this notion of quantized stability, we can support efficient deletion, since most deletions can be resolved without re-computing the centroids from scratch. Our proposed algorithm is distinct from other quantized versions of $k$-means [73], which quantize the data to minimize memory or communication costs. We present an abridged version of the algorithm here (Algorithm 1). Detailed pseudo-code for Q-$k$-means and its deletion operation may be found in Appendix B.

Q-$k$-means follows the iterative protocol as does the canonical Lloyd's algorithm (and makes use of the $k$-means++ initialization). There are four key differences from Lloyd's algorithm. First and foremost, the centroids are quantized in each iteration before updating the partition. The quantization maps each point to the nearest vertex of a uniform $\epsilon$-lattice [38]. To de-bias the quantization, we apply a random phase shift to the lattice. The particulars of the quantization scheme are discussed in Appendix B. Second, at various steps throughout the computation, we *memoize* the optimization state into the model's metadata for use at deletion time (incurring an additional $O(ktd)$ memory cost). Third, we

introduce a balance correction step, which compensates for $\gamma$-imbalanced clusters by averaging current centroids with a momentum term based on the previous centroids. Explicitly, for some $\gamma \in (0,1)$, we consider any partition $\pi_\kappa$ to be $\gamma$-imbalanced if $|\pi_\kappa| \leq \frac{\gamma n}{k}$. We may think of $\gamma$ as being the ratio of the smallest cluster size to the average cluster size. Fourth, because of the quantization, the iterations are no longer guaranteed to decrease the loss, so we have an early termination if the loss increases at any iteration. Note that the algorithm terminates almost surely.

Deletion in Q-$k$-means is straightforward. Using the metadata saved from training time, we can verify if deleting a specific datapoint would have resulted in a different *quantized centroid* than was actually computed during training. If this is the case (or if the point to be deleted is one of randomly chosen initial centroids according to $k$-means++) we must retrain from scratch to satisfy the deletion request. Otherwise, we may satisfy deletion by updating our metadata to reflect the deletion of the specified datapoint, but we do not have to recompute the centroids. Q-$k$-means directly relies the principle of quantization to enable fast deletion in expectation. It is also worth noting that Q-$k$-means also leverages on the principle of linearity to recycle computation. Since centroid computation is linear in the datapoints, it is easy to determine the centroid update due to a removal at deletion time.

---
**Algorithm 1** Quantized $k$-means (abridged)

---
**Input:** data matrix $D \in \mathbf{R}^{n \times d}$
**Parameters:** $k \in \mathbf{N}, T \in \mathbf{N}, \gamma \in (0,1), \epsilon > 0$
$c \leftarrow k^{++}(D)$ // *initialize centroids with k-means++*
Save initial centroids: save($c$).
$L \leftarrow k$-means loss of initial partition $\pi(c)$
**for** $\tau = 1$ **to** $T$ **do**
    Store current centroids: $c' \leftarrow c$
    Compute centroids: $c \leftarrow c(\pi)$
    Apply correction to $\gamma$-imbalanced partitions
    Quantize to random $\epsilon$-lattice: $\hat{c} \leftarrow Q(c; \theta)$
    Update partition: $\pi' \leftarrow \pi(\hat{c})$
    Save state to metadata: save($c, \theta, \hat{c}, |\pi'|$)
    Compute loss $L'$
    **if** $L' < L$ **then** $(c, \pi, L) \leftarrow (\hat{c}, \pi', L')$ **else break**
**end for**
**return** $c$ //output final centroids as model

---

**Deletion Time Complexity**    We turn our attention to an asymptotic time complexity analysis of Q-$k$-means deletion operation. Q-$k$-means supports deletion by quantizing the centroids, so they are stable to against small perturbations (caused by deletion of a point).

**Theorem 4.1.** *Let $D$ be a dataset on $[0,1]^d$ of size $n$. Fix parameters $T$, $k$, $\epsilon$, and $\gamma$ for Q-$k$-means. Then, Q-$k$-means supports $m$ deletions in time $O(m^2 d^{5/2}/\epsilon)$ in expectation, with probability over the randomness in the quantization phase and $k$-means++ initialization.*

The proof for the theorem is given in Appendix C. The intuition is as follows. Centroids are computed by taking an average. With enough terms in an average, the effect of a small number of those terms is negligible. The removal of those terms from the average can be interpreted as a small perturbation to the centroid. If that small perturbation is on a scale far below the granularity of the quantizing $\epsilon$-lattice, then it is unlikely to change the quantized value of the centroid. Thus, beyond stability verification, no additional computation is required for a majority of deletion requests. This result is in expectation with respect to the randomized initializations and randomized quantization phase, but is actually worst-case over all possible (normalized) dataset instances. The number of clusters $k$, iterations $T$, and cluster imbalance ratio $\gamma$ are usually small constants in many applications, and are treated as such here. Interestingly, for constant $m$ and $\epsilon$, the expected deletion time is independent of $n$ due to the stability probability increasing at the same rate as the problem size (see Appendix C). Deletion time for this method may not scale well in the high-dimensional setting. In the low-dimensional case, the most interesting interplay is between $\epsilon$, $n$, and $m$. To obtain as high-quality statistical performance as possible, it would be ideal if $\epsilon \to 0$ as $n \to \infty$. In this spirit, we can parameterize $\epsilon = n^{-\beta}$ for $\beta \in (0,1)$. We will use this parameterization for theoretical analysis of the online setting in Section 4.3.

**Theoretical Statistical Performance**    We proceed to state a theoretical guarantee on statistical performance of Q-$k$-means, which complements the asymptotic time complexity bound of the deletion operation. Recall that the loss for a $k$-means problem instance is given by the sum of squared Euclidean distance from each datapoint to its nearest centroid. Let $\mathcal{L}^*$ be the optimal loss for a particular problem instance. Achieving the optimal solution is, in general, NP-Hard [3]. Instead, we can approximate it with $k$-means++, which achieves $\mathbf{E}\mathcal{L}^{++} \leq (8\log k + 16)\mathcal{L}^*$ [5].

**Corollary 4.1.1.** *Let $\mathcal{L}$ be a random variable denoting the loss of Q-$k$-means on a particular problem instance of size $n$. Then $\mathbf{E}\mathcal{L} \leq (8\log k + 16)\mathcal{L}^* + \epsilon\sqrt{nd(8\log k + 16)\mathcal{L}^*} + \frac{1}{4}nd\epsilon^2$.*

This corollary follows from the theoretical guarantees already known to apply to Lloyd's algorithm when initialized with $k$-means++, given by [5]. The proof can be found in Appendix C. We can

interpret the bound by looking at the ratio of expected loss upper bounds for $k$-means++ and Q-$k$-means. If we assume our problem instance is generated by iid samples from some arbitrary non-atomic distribution, then it follows that $\mathcal{L}^* = O(n)$. Taking the loss ratio of upper bounds yields $\mathbf{E}\mathcal{L}/\mathbf{E}\mathcal{L}^{++} \leq 1 + O(d\epsilon^2 + \sqrt{d}\epsilon)$. Ensuring that $\epsilon << 1/\sqrt{d}$ implies the upper bound is as good as that of $k$-means++.

## 4.2 Divide-and-Conquer $k$-Means

We turn our attention to another variant of Lloyd's algorithm that also supports efficient deletion, albeit through quite different means. We refer to this algorithm as Divide-and-Conquer $k$-means (DC-$k$-means). At a high-level, DC-$k$-means works by partitioning the dataset into small sub-problems, solving each sub-problem as an independent $k$-means instance, and recursively merging the results. We present pseudo-code for DC-$k$-means here, and we refer the reader to Appendix B for pseudo-code of the deletion operation.

DC-$k$-means operates on a perfect $w$-ary tree of height $h$ (this could be relaxed to any rooted tree). The original dataset is *partitioned* into each leaf in the tree as a uniform multinomial random variable with datapoints as trials and leaves as outcomes. At each of these leaves, we solve for some number of centroids via $k$-means++. When we merge leaves into their parent node, we construct a new dataset consisting of all the centroids from each leaf. Then, we compute new centroids at the parent via another instance of $k$-means++. For simplicity, we keep $k$ fixed throughout all of the sub-problems in the tree, but this could be relaxed. We make use of the tree hierarchy to *modularize* the computation's dependence on the data. At deletion time, we need only to recompute the sub-problems from *one* leaf up to the root. This observation allows us to support fast deletion operations.

---
**Algorithm 2** DC-$k$-means

**Input:** data matrix $D \in \mathbf{R}^{n \times d}$
**Parameters:** $k \in \mathbf{N}$, $T \in \mathbf{N}$, tree width $w \in \mathbf{N}$, tree height $h \in \mathbf{N}$
Initialize a $w$-ary tree of height $h$ such that each node has a pointer to a dataset and centroids
**for** $i = 1$ **to** $n$ **do**
    Select a leaf node uniformly at random
    node.dataset.add($D_i$)
**end for**
**for** $l = h$ **down to** $0$ **do**
    **for** each node in level $l$ **do**
        $c \leftarrow$ k-means++(node.dataset,$k$,$T$)
        node.centroids $\leftarrow c$
        **if** $l > 0$ **then**
            node.parent.dataset.add($c$)
        **else**
            **save** all nodes as metadata
            **return** $c$ //model output
        **end if**
    **end for**
**end for**

---

Our method has close similarities to pre-existing distributed $k$-means algorithms [69, 67, 9, 7, 39, 8, 92], but is in fact distinct (not only in that it is modified for deletion, but also in that it operates over general rooted trees). For simplicity, we restrict our discussion to only the simplest of divide-and-conquer trees. We focus on depth-1 trees with $w$ leaves where each leaf solves for $k$ centroids. This requires only one merge step with a root problem size of $kn/w$.

Analogous to how $\epsilon$ serves as a knob to trade-off between deletion efficiency and statistical performance in Q-$k$-means, for DC-$k$-means, we imagine that $w$ might also serve as a similar knob. For example, if $w = 1$, DC-$k$-means degenerates into canonical Lloyd's (as does Q-$k$-means as $\epsilon \to 0$). The dependence of statistical performance on tree width $w$ is less theoretically tractable than that of Q-$k$-means on $\epsilon$, but in Appendix D, we empirically show that statistical performance tends to decrease as $w$ increases, which is perhaps somewhat expected.

As we show in our experiments, depth-1 DC-$k$-means demonstrates an empirically compelling trade-off between deletion time and statistical performance. There are various other potential extensions of this algorithm, such as weighting centroids based on cluster mass as they propagate up the tree or exploring the statistical performance of deeper trees.

**Deletion Time Complexity** For ensuing asymptotic analysis, we may consider parameterizing tree width $w$ as $w = \Theta(n^\rho)$ for $\rho \in (0,1)$. As before, we treat $k$ and $T$ as small constants. Although intuitive, there are some technical minutia to account for to prove correctness and runtime for the DC-$k$-means deletion operation. The proof of Proposition 3.2 may be found in Appendix C.

**Proposition 4.2.** *Let $D$ be a dataset on $\mathbf{R}^d$ of size $n$. Fix parameters $T$ and $k$ for DC-$k$-means. Let $w = \Theta(n^\rho)$ and $\rho \in (0,1)$ Then, with a depth-1, $w$-ary divide-and-conquer tree, DC-$k$-means supports $m$ deletions in time $O(m\mathbf{max}\{n^\rho, n^{1-\rho}\}d)$ in expectation with probability over the randomness in dataset partitioning.*

### 4.3 Amortized Runtime Complexity in Online Deletion Setting

We state the amortized computation time for both of our algorithms in the online deletion setting defined in Section 3. We are in an asymptotic regime where the number of deletions $m = \Theta(n^\alpha)$ for $0 < \alpha < 1$ (see Appendix C for more details). Recall the $\Omega(\frac{n}{m})$ lower bound from Section 3. For a particular fractional power $\alpha$, an algorithm achieving the optimal asymptotic lower bound on amortized computation is said to be $\alpha$-*deletion efficient*. This corresponds to achieving an amortized runtime of $O(n^{1-\alpha})$. The following corollaries result from direct calculations which may be found in Appendix C. Note that Corollary 4.2.2 assumes DC-$k$-means is training sequentially.

**Corollary 4.2.1.** *With $\epsilon = \Theta(n^{-\beta})$, for $0 < \beta < 1$, the Q-k-means algorithm is $\alpha$-deletion efficient in expectation if $\alpha \leq \frac{1-\beta}{2}$.*

**Corollary 4.2.2.** *With $w = \Theta(n^\rho)$, for $0 < \rho < 1$, and a depth-1 $w$-ary divide-and-conquer tree, DC-k-means is $\alpha$-deletion efficient in expectation if $\alpha < 1 - \textbf{max}\{1-\rho, \rho\}$.*

## 5 Experiments

With a theoretical understanding in hand, we seek to empirically characterize the trade-off between runtime and performance for the proposed algorithms. In this section, we provide proof-of-concept for our algorithms by benchmarking their amortized runtimes and clustering quality on a simulated stream of online deletion requests. As a baseline, we use the canonical Lloyd's algorithm initialized by $k$-means++ seeding [53, 5]. Following the broader literature, we refer to this baseline simply as $k$-means, and refer to our two proposed methods as Q-$k$-means and DC-$k$-means.

**Datasets**    We run our experiments on five real, publicly available datasets: `Celltype` ($N = 12,009$, $D = 10$, $K = 4$) [42], `Covtype` ($N = 15,120$, $D = 52$, $K = 7$) [12], `MNIST` ($N = 60,000$, $D = 784$, $K = 10$) [51], `Postures` ($N = 74,975$, $D = 15$, $K = 5$) [35, 34] , `Botnet` ($N = 1,018,298$, $D = 115$, $K = 11$)[56], and a synthetic dataset made from a Gaussian mixture model which we call `Gaussian` ($N = 100,000$, $D = 25$, $K = 5$). We refer the reader to Appendix D for more details on the datasets. All datasets come with ground-truth labels as well. Although we do not make use of the labels at learning time, we can use them to evaluate the statistical quality of the clustering methods.

**Online Deletion Benchmark**    We simulate a stream of 1,000 deletion requests, selected uniformly at random and without replacement. An algorithm trains once, on the full dataset, and then runs its deletion operation to satisfy each request in the stream, producing an intermediate model at each request. For the canonical $k$-means baseline, deletions are satisfied by re-training from scratch.

**Protocol**    To measure statistical performance, we evaluate with three metrics (see Section 5.1) that measure cluster quality. To measure deletion efficiency, we measure the wall-clock time to complete our online deletion benchmark. For both of our proposed algorithms, we always fix 10 iterations of Lloyd's, and all other parameters are selected with simple but effective heuristics (see Appendix D). This alleviates the need to tune them. To set a fair $k$-means baseline, when reporting runtime on the online deletion benchmark, we also fix 10 iterations of Lloyd's, but when reporting statistical performance metrics, we run until convergence. We run five replicates for each method on each dataset and include standard deviations with all our results. We refer the reader to Appendix D for more experimental details.

### 5.1 Statistical Performance Metrics

To evaluate clustering performance of our algorithms, the most obvious metric is the optimization loss of the $k$-means objective. Recall that this is the sum of square Euclidean distances from each datapoint to its nearest centroid. To thoroughly validate the statistical performance of our proposed algorithms, we additionally include two canonical clustering performance metrics.

**Silhouette Coefficient** [72]: This coefficient measures a type of correlation (between -1 and +1) that captures how dense each cluster is and how well-separated different clusters are. The silhouette coefficient is computed without ground-truth labels, and uses only spatial information. Higher scores indicate denser, more well-separated clusters.

**Normalized Mutual Information (NMI)** [87, 49]: This quantity measures the agreement of the assigned clusters to the ground-truth labels, up to permutation. NMI is upper bounded by 1, achieved by perfect assignments. Higher scores indicate better agreement between clusters and ground-truth labels.

## 5.2 Summary of Results

We summarize our key findings in four tables. In Tables 1-3, we report the statistical clustering performance of the 3 algorithms on each of the 6 datasets. In Table 1, we report the optimization loss ratios of our proposed methods over the $k$-means++ baseline.

In Table 2, we report the silhouette coefficient for the clusters. In Table 3, we report the NMI. In Table 4, we report the amortized total runtime of training and deletion for each method. **Overall, we see that the statistical clustering performance of the three methods are competitive.**

**Furthermore, we find that both proposed algorithms yield orders of magnitude of speedup.** As expected from the theoretical analysis, Q-$k$-means offers greater speed-ups when the dimension is lower relative to the sample size, whereas DC-$k$-means is more consistent across dimensionalities.

Table 1: Loss Ratio

| Dataset | $k$-means | Q-$k$-means | DC-$k$-means |
|---|---|---|---|
| **Celltype** | 1.0±0.0 | 1.158±0.099 | 1.439±0.157 |
| **Covtype** | 1.0±0.029 | 1.033±0.017 | 1.017±0.031 |
| **MNIST** | 1.0±0.002 | 1.11±0.004 | 1.014±0.003 |
| **Postures** | 1.0±0.004 | 1.014±0.015 | 1.034±0.017 |
| **Gaussian** | 1.0±0.014 | 1.019±0.019 | 1.003±0.014 |
| **Botnet** | 1.0±0.126 | 1.018±0.014 | 1.118±0.102 |

Table 2: Silhouette Coefficients (higher is better)

| Dataset | $k$-means | Q-$k$-means | DC-$k$-means |
|---|---|---|---|
| **Celltype** | 0.384±0.001 | 0.367±0.048 | 0.422±0.057 |
| **Covtype** | 0.238±0.027 | 0.203±0.026 | 0.222±0.017 |
| **Gaussian** | 0.036±0.002 | 0.031±0.002 | 0.035±0.001 |
| **Postures** | 0.107±0.003 | 0.107±0.004 | 0.109±0.005 |
| **Gaussian** | 0.066±0.007 | 0.053±0.003 | 0.071±0.004 |
| **Botnet** | 0.583±0.042 | 0.639±0.028 | 0.627±0.046 |

Table 3: Normalized Mutual Information (higher is better)

| Dataset | $k$-means | Q-$k$-means | DC-$k$-means |
|---|---|---|---|
| **Celltype** | 0.36±0.0 | 0.336±0.032 | 0.294±0.067 |
| **Covtype** | 0.311±0.009 | 0.332±0.024 | 0.335±0.02 |
| **MNIST** | 0.494±0.006 | 0.459±0.011 | 0.494±0.004 |
| **Gaussian** | 0.319±0.024 | 0.245±0.024 | 0.318±0.024 |
| **Postures** | 0.163±0.018 | 0.169±0.012 | 0.173±0.011 |
| **Botnet** | 0.708±0.048 | 0.73±0.015 | 0.705±0.039 |

Table 4: Amortized Runtime in Online Deletion Benchmark (Train once + 1,000 Deletions)

| Dataset | $k$-means Runtime (s) | Q-$k$-means Runtime (s) | Q-$k$-means Speedup | DC-$k$-means Runtime (s) | DC-$k$-means Speedup |
|---|---|---|---|---|---|
| **Celltype** | 4.241±0.248 | 0.026±0.011 | 163.286× | 0.272±0.007 | 15.6× |
| **Covtype** | 6.114±0.216 | 0.454±0.276 | 13.464× | 0.469±0.021 | 13.048× |
| **MNIST** | 65.038±1.528 | 29.386±0.728 | 2.213× | 2.562±0.056 | 25.381× |
| **Postures** | 26.616±1.222 | 0.413±0.305 | 64.441× | 1.17±0.398 | 22.757× |
| **Gaussian** | 206.631±67.285 | 0.393±0.104 | 525.63× | 5.992±0.269 | 34.483× |
| **Botnet** | 607.784±64.687 | 1.04±0.368 | 584.416× | 8.568±0.652 | 70.939× |

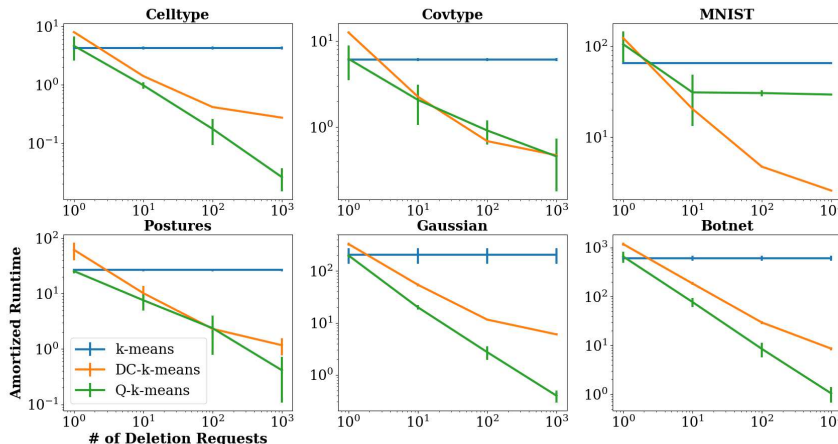

Figure 1: Online deletion efficiency: # of deletions vs. amortized runtime (secs) for 3 algorithms on 6 datasets.

In particular, note that `MNIST` has the highest $d/n$ ratio of the datasets we tried, followed by `Covtype`, These two datasets are, respectively, the datasets for which Q-$k$-means offers the least speedup. On the other hand, DC-$k$-means offers consistently increasing speedup as $n$ increases, for fixed $d$. Furthermore, we see that Q-$k$-means tends to have higher variance around its deletion efficiency, due to the randomness in centroid stabilization having a larger impact than the randomness in the dataset partitioning. We remark that 1,000 deletions is less than 10% of every dataset we test on, and statistical performance remains virtually unchanged throughout the benchmark. In Figure 1, we plot the amortized runtime on the online deletion benchmark as a function of number of deletions in the stream. We refer the reader to Appendix D for supplementary experiments providing more detail on our methods.

# 6   Discussion

At present, the main options for deletion efficient supervised methods are linear models, support vector machines, and non-parametric regressions. While our analysis here focuses on the concrete problem of clustering, we have proposed four design principles which we envision as the pillars of deletion efficient learning algorithms. We discuss the potential application of these methods to other supervised learning techniques.

**Segmented Regression**   Segmented (or piece-wise) linear regression is a common relaxation of canonical regression models [58, 59, 57]. It should be possible to support a variant of segmented regression by combining Q-$k$-means with linear least squares regression. Each cluster could be given a separate linear model, trained only on the datapoints in said cluster. At deletion time, Q-$k$-means would likely keep the clusters stable, enabling a simple linear update to the model corresponding to the cluster from which the deleted point belonged.

**Kernel Regression**   Kernel regressions in the style of random Fourier features [70] could be readily amended to support efficient deletions for large-scale supervised learning. Random features do not depend on data, and thus only the linear layer over the feature space requires updating for deletion. Furthermore, random Fourier feature methods have been shown to have affinity for quantization [91].

**Decision Trees and Random Forests**   Quantization is also a promising approach for decision trees. By quantizing or randomizing decision tree splitting criteria (such as in [36]) it seems possible to support efficient deletion. Furthermore, random forests have a natural affinity with bagging, which naturally can be used to impose modularity.

**Deep Neural Networks and Stochastic Gradient Descent**   A line of research has observed the robustness of neural network training robustness to quantization and pruning [84, 46, 40, 71, 25, 52]. It could be possible to leverage these techniques to quantize gradient updates during SGD-style optimization, enabling a notion of parameter stability analgous to that in Q-$k$-means. This would require larger batch sizes and fewer gradient steps in order to scale well. It is also possible that approximate deletion methods may be able to overcome shortcomings of exact deletion methods for large neural models.

# 7   Conclusion

In this work, we developed a notion of deletion efficiency for large-scale learning systems, proposed provably deletion efficient unsupervised clustering algorithms, and identified potential algorithmic principles that may enable deletion efficiency for other learning algorithms and paradigms. We have only scratched the surface of understanding deletion efficiency in learning systems. Throughout, we made a number of simplifying assumptions, such that there is only one model and only one database in our system. We also assumed that user-based deletion requests correspond to only a single data point. Understanding deletion efficiency in a system with many models and many databases, as well as complex user-to-data relationships, is an important direction for future work.

**Acknowledgments:**   This research was partially supported by NSF Awards AF:1813049, CCF:1704417, and CCF 1763191, NIH R21 MD012867-01, NIH P30AG059307, an Office of Naval Research Young Investigator Award (N00014-18-1-2295), a seed grant from Stanford's Institute for Human-Centered AI, and the Chan-Zuckerberg Initiative. We would also like to thank I. Lemhadri, B. He, V. Bagaria, J. Thomas and anonymous reviewers for helpful discussion and feedback.

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
