[Supplementary Material]

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

# A  Supplementary Materials

Here we provide material supplementary to the main text. While some of the material provided here may be somewhat redundant, it also contains technical minutia perhaps too detailed for the main body.

## A.1  Online Data Deletion

We precisely define the notion of a *learning algorithm* for theoretical discussion in the context of data deletion.

**Definition A.1.**  Learning Algorithm

A *learning* algorithm $A$ is an algorithm (on some standard model of computation) taking values in some hypothesis space and metadata space $\mathcal{H} \times \mathcal{M}$ based on an input dataset $D$. Learning algorithm $A$ may be randomized, implying a conditional distribution over $\mathcal{H} \times \mathcal{M}$ given $D$. Finally, learning algorithms must process each datapoint in $D$ at least once, and are constrained to sequential computation only, yielding a runtime bounded by $\Omega(n)$.

We re-state the definition of data deletion. We distinguish between a *deletion operation* and a *robust deletion operation*. We focus on the former throughout our main body, as it is appropriate for average-case analysis in a non-security context. We use $=_d$ to denote distributional equality.

**Definition A.2.**  Data Deletion Operation

Fix any dataset $D$ and learning algorithm $A$. Operation $R_A$ is a *deletion operation* for $A$ if $R_A(D, A(D), i) =_d A(D_{-i})$ for any $i$ selected independently of $A(D)$.

For notational simplicity, we may let $R_A$ refer to an entire sequence of deletions ($\Delta = \{i_1, i_2, ..., i_m\}$) by writing $R_A(D, A(D), \Delta)$. This notation means the output of a sequence of applications of $R_A$ to each $i$ in deletion sequence $\Delta$. We also may drop the dependence on $A$ when it is understood for which $A$ the deletion operation $R$ corresponds. We also drop the arguments for $A$ and $R$ when they are understood from context. For example, when dataset $D$ can be inferred from context, we let $A_{-i}$ directly mean $A(D_{-i})$ and when and deletion stream $\Delta$ can be inferred, we let $R$ directly mean $R(D, A(D), \Delta)$.

Our definition is somewhat analogous to *information-theoretic* (or perfect) secrecy in cryptography [78]. Much like in cryptography, it is possible to relax to weaker notions – for example, by statistically approximating deletion and bounding the amount of computation some hypothetical adversary could use to determine if a genuine deletion took place. Such relaxations are required for encryption algorithms because perfect secrecy can only be achieved via one-time pad [78]. In the context of deletion operations, retraining from scratch is, at least slightly, analogous to one-time pad encryption: both are simple solutions that satisfy distributional equality requirements, but both solutions are impractical. However, unlike in encryption, when it comes to deletion, we can, in fact, at least for some learning algorithms, find deletion operations that would be both practical and perfect.

The upcoming *robust* definition may be of more interest in a worst-case, security setting. In such a setting, an adaptive adversary makes deletion requests while also having perfect eavesdropping capabilities to the server (or at least the internal state of the learning algorithm, model and metadata).

**Definition A.3.**  Robust Data Deletion Operation

Fix any dataset $D$ and learning algorithm $A$. Operation $R_A$ is a *robust deletion operation* if $R_A(D, A(D), i) =_d A(D_{-i})$ in distribution, for any $i$, perhaps selected by an adversarial agent with knowledge of $A(D)$.

To illustrate the difference between these two definitions, consider Q-$k$-means and DC-$k$-means. Assume an adversary has compromised the server with read-access and gained knowledge of the algorithm's internal state. Further assume that said adversary may issue deletion requests. Such a powerful adversary could compromise the exactness of DC-$k$-means deletion operations by deleting datapoints from *specific* leaves. For example, if the adversary always deletes datapoints partitioned to the first leaf, then the number of datapoints assigned to each leaf is no longer uniform or independent of deletion requests. In principle, this, at least rigorously speaking, violates equality in distribution. Note that this can only occur if requests are somehow dependent on the partition. However, despite an adversary being able to compromise the correctness of the deletion operation, it cannot compromise the efficiency. That is because efficiency depends on the maximum number of datapoints partitioned to a particular leaf, a quantity which is decided randomly without input from the adversary.

In the case of Q-$k$-means we can easily see the deletion is robust to the adversary by the enforced equality of outcome imposed by the deletion operation. However, an adversary with knowledge of algorithm state could make the Q-$k$-means deletion operation entirely inefficient by always deleting an initial centroid. This causes every single deletion to be satisfied by retraining from scratch. From the security perspective, it could be of interest to study deletion operations that are both robust and efficient.

We continue by defining the *online* data deletion setting in the *average-case*.

**Definition A.4.** Online Data Deletion (Average-Case)

We may formally define the runtime in the online deletion setting as the expected runtime of Algorithm 3. We amortize the total runtime by $m$.

---

**Algorithm 3** Online Data Deletion

---

**Input:** Dataset $D$, learning algorithm $A$, deletion operation $R$
**Parameters:** $\alpha \in (0,1)$
$\mu \leftarrow A(D)$
$m \leftarrow \Theta(|D|^{\alpha})$
**for** $\tau = 1$ **to** $m$ **do**
   $i \leftarrow \mathbf{Unif}[1,...,|D|]$ //constant time
   $\mu \leftarrow R(D,\mu,i)$
   $D \leftarrow D_{-i}$ //constant time
**end for**

---

**Fractional power regime.** For dataset size $n$, when $m = \Theta(n^{\alpha})$ for $0 < \alpha < 1$, we say we are in the *fractional power regime*. For $k$-means, our proposed algorithms achieve the ideal lower bound for small enough $\alpha$, but not for all $\alpha$ in $(0,1)$.

Online data deletion is interesting in both the *average-case* setting (treated here), where the indices $\{i_1,...,i_m\}$ are chosen uniformly and independently without replacement, as well as in a *worst-case* setting, where the sequence of indices is computed adversarially (left for future work). It may also be practical to include a bound on the amount of memory available to the data deletion operation and model (including metadata) as an additional constraint.

**Definition A.5.** Deletion Efficient Learning Algorithm

Recall the $\Omega(n/m)$ lower bound on amortized computation for any sequential learning algorithm in the online deletion setting (Section 2). Given some fractional power scaling $m = \Theta(n^{\alpha})$, we say an algorithm $A$ is $\alpha$-deletion efficient if it runs Algorithm 3 in amortized time $O(n^{1-\alpha})$.

**Inference Time**    Of course, lazy learning and non-parametric techniques are a clear exception to our notions of learning algorithm. For these methods, data is processed at inference time rather than training time – a more complete study of the systems trade-offs between training time, inference time, and deletion time is left for future work.

## A.2   Approximate Data Deletion

We present one possible relaxation from exact to *approximate* data deletion.

**Definition A.6.** Approximate deletion

We say that such a data deletion operation $R_A$ is an $\delta$-deletion for algorithm $A$ if, for all $D$ and for every measurable subset $S \subseteq \mathcal{H} \times M$:

$$\mathbf{Pr}[A(D_{-i}) \in S | D_{-i}] \geq \delta \mathbf{Pr}[R_A(D,A(D),i) \in S | D_{-i}]$$

The above definition corresponds to requiring that the probability that the data deletion operation returns a model in some specified set, $S$, cannot be more than a $\delta^{-1}$ factor larger than the probability that algorithm $A$ retrained on the dataset $D_{-i}$ returns a model in that set. We note that the above definition *does* allow for the possibility that some outcomes that have positive probability under $A(D_{-i})$ have zero probability under the deletion operation. In such a case, an observer could conclude that the model was returned by running $A$ from scratch.

### A.2.1 Approximate Deletion and Differential Privacy

We recall the definition of differential privacy [30]. A map, $A$, from a dataset $D$ to a set of outputs, $\mathcal{H}$, is said to be $\epsilon$-differentially private if, for any two datasets $D_1, D_2$ that differ in a single datapoint, and any subset $S \subset \mathcal{H}$, $\mathbf{Pr}[A(D_1) \in S] \leq e^\epsilon \cdot \mathbf{Pr}[A(D_2) \in S]$.

Under the relaxed notion of data deletion, it is natural to consider privatization as a manner to support approximation deletion. The idea could be to privatize the model, and then resolve deletion requests by ignoring them. However, there are some nuances involved here that one should be careful of. For example, differential privacy does not privatize the number of datapoints, but this should not be leaked in data deletion. Furthermore, since we wish to support a stream of deletions in the online setting, we would need to use *group differential privacy* [30], which can greatly increase the amount of noise needed for privatization. Even worse, this requires selecting the group size (i.e. total privacy budget) during training time (at least for canonical constructions such as the Laplace mechanism). In differential privacy, this group size is not necessarily a hidden parameter. In the context of deletion, it could leak information about the total dataset size as well as how many deletions any given model instance has processed. While privatization-like methods are perhaps a viable approach to support approximate deletion, there remain some technical details to work out, and this is left for future work.

## B  Algorithmic Details

In Appendix B, we present psuedo-code for the algorithms described in Section 3. We also reference `https://github.com/tginart/deletion-efficient-kmeans` for Python implementations of our algorithms.

### B.1  Quantized $k$-Means

We present the psuedo-code for Q-$k$-means (Algo. 4). Q-$k$-means follows the iterative protocol as the canonical Lloyd's (and makes use of the $k$-means++ initialization). As mentioned in the main body, there are four key variations from the canonical Lloyd's algorithm that make this method different: quantization, memoization, balance correction, and early termination. The memoization of the optimization state and the early termination for increasing loss are self-explanatory from Algo. 4. We provide more details concerning the quantization step and the balance correction in Appendix B.1.1 and B.1.2 respectively.

---

**Algorithm 4** Quantized $k$-means

---

**Input:** data matrix $D \in \mathbf{R}^{n \times d}$
**Parameters:** $k \in \mathbf{N}, T \in \mathbf{N}, \gamma \in (0,1), \epsilon > 0$
$c \leftarrow k^{++}(D)$ // *initialize centroids with k-means++*
Save initial centroids: $\mathsf{save}(c)$.
$L \leftarrow k$-means loss of initial partition $\pi$
**for** $\tau = 1$ **to** $T$ **do**
    Store current centroids: $c' \leftarrow c$
    Compute centroids: $c \leftarrow c(\pi)$
    **for** $\kappa = 1$ **to** $k$ **do**
        **if** $|\pi(c_\kappa)| < \frac{\gamma n}{k}$  **then**
            Apply correction to $\gamma$-imbalanced partition:
            $c_\kappa \leftarrow |\pi(c_\kappa)|c_\kappa + (\frac{\gamma n}{k} - |\pi(c_\kappa)|)c'_\kappa$
        **end if**
    **end for**
    Generate random phase $\theta \sim \mathbf{Unif}[-\frac{1}{2}, \frac{1}{2}]^d$
    Quantize to $\epsilon$-lattice: $\hat{c} \leftarrow Q(c; \theta)$
    Update partition: $\pi' \leftarrow \pi(\hat{c})$
    Save state to metadata: $\mathsf{save}(c, \theta, \hat{c}, |\pi'|)$
    Compute loss $L'$
    **if** $L' < L$ **then**
        $(c, \pi, L) \leftarrow (\hat{c}, \pi', L')$ //update state
    **else**
        **break**
    **end if**
**end for**
**return** $c$ //output final centroids as model

---

# A  Supplementary Materials

Here we provide material supplementary to the main text. While some of the material provided here may be somewhat redundant, it also contains technical minutia perhaps too detailed for the main body.

## A.1  Online Data Deletion

We precisely define the notion of a *learning algorithm* for theoretical discussion in the context of data deletion.

**Definition A.1.**  Learning Algorithm

A *learning* algorithm $A$ is an algorithm (on some standard model of computation) taking values in some hypothesis space and metadata space $\mathcal{H} \times \mathcal{M}$ based on an input dataset $D$. Learning algorithm $A$ may be randomized, implying a conditional distribution over $\mathcal{H} \times \mathcal{M}$ given $D$. Finally, learning algorithms must process each datapoint in $D$ at least once, and are constrained to sequential computation only, yielding a runtime bounded by $\Omega(n)$.

We re-state the definition of data deletion. We distinguish between a *deletion operation* and a *robust deletion operation*. We focus on the former throughout our main body, as it is appropriate for average-case analysis in a non-security context. We use $=_d$ to denote distributional equality.

**Definition A.2.**  Data Deletion Operation

Fix any dataset $D$ and learning algorithm $A$. Operation $R_A$ is a *deletion operation* for $A$ if $R_A(D, A(D), i) =_d A(D_{-i})$ for any $i$ selected independently of $A(D)$.

For notational simplicity, we may let $R_A$ refer to an entire sequence of deletions ($\Delta = \{i_1, i_2, ..., i_m\}$) by writing $R_A(D, A(D), \Delta)$. This notation means the output of a sequence of applications of $R_A$ to each $i$ in deletion sequence $\Delta$. We also may drop the dependence on $A$ when it is understood for which $A$ the deletion operation $R$ corresponds. We also drop the arguments for $A$ and $R$ when they are understood from context. For example, when dataset $D$ can be inferred from context, we let $A_{-i}$ directly mean $A(D_{-i})$ and when and deletion stream $\Delta$ can be inferred, we let $R$ directly mean $R(D, A(D), \Delta)$.

Our definition is somewhat analogous to *information-theoretic* (or perfect) secrecy in cryptography [78]. Much like in cryptography, it is possible to relax to weaker notions – for example, by statistically approximating deletion and bounding the amount of computation some hypothetical adversary could use to determine if a genuine deletion took place. Such relaxations are required for encryption algorithms because perfect secrecy can only be achieved via one-time pad [78]. In the context of deletion operations, retraining from scratch is, at least slightly, analogous to one-time pad encryption: both are simple solutions that satisfy distributional equality requirements, but both solutions are impractical. However, unlike in encryption, when it comes to deletion, we can, in fact, at least for some learning algorithms, find deletion operations that would be both practical and perfect.

The upcoming *robust* definition may be of more interest in a worst-case, security setting. In such a setting, an adaptive adversary makes deletion requests while also having perfect eavesdropping capabilities to the server (or at least the internal state of the learning algorithm, model and metadata).

**Definition A.3.**  Robust Data Deletion Operation

Fix any dataset $D$ and learning algorithm $A$. Operation $R_A$ is a *robust deletion operation* if $R_A(D, A(D), i) =_d A(D_{-i})$ in distribution, for any $i$, perhaps selected by an adversarial agent with knowledge of $A(D)$.

To illustrate the difference between these two definitions, consider Q-$k$-means and DC-$k$-means. Assume an adversary has compromised the server with read-access and gained knowledge of the algorithm's internal state. Further assume that said adversary may issue deletion requests. Such a powerful adversary could compromise the exactness of DC-$k$-means deletion operations by deleting datapoints from *specific* leaves. For example, if the adversary always deletes datapoints partitioned to the first leaf, then the number of datapoints assigned to each leaf is no longer uniform or independent of deletion requests. In principle, this, at least rigorously speaking, violates equality in distribution. Note that this can only occur if requests are somehow dependent on the partition. However, despite an adversary being able to compromise the correctness of the deletion operation, it cannot compromise the efficiency. That is because efficiency depends on the maximum number of datapoints partitioned to a particular leaf, a quantity which is decided randomly without input from the adversary.

In the case of Q-$k$-means we can easily see the deletion is robust to the adversary by the enforced equality of outcome imposed by the deletion operation. However, an adversary with knowledge of algorithm state could make the Q-$k$-means deletion operation entirely inefficient by always deleting an initial centroid. This causes every single deletion to be satisfied by retraining from scratch. From the security perspective, it could be of interest to study deletion operations that are both robust and efficient.

We continue by defining the *online* data deletion setting in the *average-case*.

**Definition A.4.** Online Data Deletion (Average-Case)

We may formally define the runtime in the online deletion setting as the expected runtime of Algorithm 3. We amortize the total runtime by $m$.

---

**Algorithm 3** Online Data Deletion

---

**Input:** Dataset $D$, learning algorithm $A$, deletion operation $R$
**Parameters:** $\alpha \in (0,1)$
$\mu \leftarrow A(D)$
$m \leftarrow \Theta(|D|^{\alpha})$
**for** $\tau = 1$ **to** $m$ **do**
    $i \leftarrow \mathbf{Unif}[1,...,|D|]$ //constant time
    $\mu \leftarrow R(D,\mu,i)$
    $D \leftarrow D_{-i}$ //constant time
**end for**

---

**Fractional power regime.** For dataset size $n$, when $m = \Theta(n^{\alpha})$ for $0 < \alpha < 1$, we say we are in the *fractional power regime*. For $k$-means, our proposed algorithms achieve the ideal lower bound for small enough $\alpha$, but not for all $\alpha$ in $(0,1)$.

Online data deletion is interesting in both the *average-case* setting (treated here), where the indices $\{i_1,...,i_m\}$ are chosen uniformly and independently without replacement, as well as in a *worst-case* setting, where the sequence of indices is computed adversarially (left for future work). It may also be practical to include a bound on the amount of memory available to the data deletion operation and model (including metadata) as an additional constraint.

**Definition A.5.** Deletion Efficient Learning Algorithm

Recall the $\Omega(n/m)$ lower bound on amortized computation for any sequential learning algorithm in the online deletion setting (Section 2). Given some fractional power scaling $m = \Theta(n^{\alpha})$, we say an algorithm $A$ is $\alpha$-deletion efficient if it runs Algorithm 3 in amortized time $O(n^{1-\alpha})$.

**Inference Time**   Of course, lazy learning and non-parametric techniques are a clear exception to our notions of learning algorithm. For these methods, data is processed at inference time rather than training time – a more complete study of the systems trade-offs between training time, inference time, and deletion time is left for future work.

## A.2 Approximate Data Deletion

We present one possible relaxation from exact to *approximate* data deletion.

**Definition A.6.** Approximate deletion

We say that such a data deletion operation $R_A$ is an $\delta$-deletion for algorithm $A$ if, for all $D$ and for every measurable subset $S \subseteq \mathcal{H} \times M$:

$$\mathbf{Pr}[A(D_{-i}) \in S | D_{-i}] \geq \delta \mathbf{Pr}[R_A(D,A(D),i) \in S | D_{-i}]$$

The above definition corresponds to requiring that the probability that the data deletion operation returns a model in some specified set, $S$, cannot be more than a $\delta^{-1}$ factor larger than the probability that algorithm $A$ retrained on the dataset $D_{-i}$ returns a model in that set. We note that the above definition *does* allow for the possibility that some outcomes that have positive probability under $A(D_{-i})$ have zero probability under the deletion operation. In such a case, an observer could conclude that the model was returned by running $A$ from scratch.

### A.2.1 Approximate Deletion and Differential Privacy

We recall the definition of differential privacy [30]. A map, $A$, from a dataset $D$ to a set of outputs, $\mathcal{H}$, is said to be $\epsilon$-differentially private if, for any two datasets $D_1, D_2$ that differ in a single datapoint, and any subset $S \subset \mathcal{H}$, $\mathbf{Pr}[A(D_1) \in S] \leq e^{\epsilon} \cdot \mathbf{Pr}[A(D_2) \in S]$.

Under the relaxed notion of data deletion, it is natural to consider privatization as a manner to support approximation deletion. The idea could be to privatize the model, and then resolve deletion requests by ignoring them. However, there are some nuances involved here that one should be careful of. For example, differential privacy does not privatize the number of datapoints, but this should not be leaked in data deletion. Furthermore, since we wish to support a stream of deletions in the online setting, we would need to use *group differential privacy* [30], which can greatly increase the amount of noise needed for privatization. Even worse, this requires selecting the group size (i.e. total privacy budget) during training time (at least for canonical constructions such as the Laplace mechanism). In differential privacy, this group size is not necessarily a hidden parameter. In the context of deletion, it could leak information about the total dataset size as well as how many deletions any given model instance has processed. While privatization-like methods are perhaps a viable approach to support approximate deletion, there remain some technical details to work out, and this is left for future work.

## B  Algorithmic Details

In Appendix B, we present psuedo-code for the algorithms described in Section 3. We also reference `https://github.com/tginart/deletion-efficient-kmeans` for Python implementations of our algorithms.

### B.1  Quantized $k$-Means

We present the psuedo-code for Q-$k$-means (Algo. 4). Q-$k$-means follows the iterative protocol as the canonical Lloyd's (and makes use of the $k$-means++ initialization). As mentioned in the main body, there are four key variations from the canonical Lloyd's algorithm that make this method different: quantization, memoization, balance correction, and early termination. The memoization of the optimization state and the early termination for increasing loss are self-explanatory from Algo. 4. We provide more details concerning the quantization step and the balance correction in Appendix B.1.1 and B.1.2 respectively.

---

**Algorithm 4** Quantized $k$-means

**Input:** data matrix $D \in \mathbf{R}^{n \times d}$
**Parameters:** $k \in \mathbf{N}, T \in \mathbf{N}, \gamma \in (0,1), \epsilon > 0$
$c \leftarrow k^{++}(D)$ // *initialize centroids with k-means++*
Save initial centroids: $\mathsf{save}(c)$.
$L \leftarrow k$-means loss of initial partition $\pi$
**for** $\tau = 1$ **to** $T$ **do**
    Store current centroids: $c' \leftarrow c$
    Compute centroids: $c \leftarrow c(\pi)$
    **for** $\kappa = 1$ **to** $k$ **do**
        **if** $|\pi(c_\kappa)| < \frac{\gamma n}{k}$ **then**
            Apply correction to $\gamma$-imbalanced partition:
            $c_\kappa \leftarrow |\pi(c_\kappa)| c_\kappa + (\frac{\gamma n}{k} - |\pi(c_\kappa)|) c'_\kappa$
        **end if**
    **end for**
    Generate random phase $\theta \sim \mathbf{Unif}[-\frac{1}{2}, \frac{1}{2}]^d$
    Quantize to $\epsilon$-lattice: $\hat{c} \leftarrow Q(c; \theta)$
    Update partition: $\pi' \leftarrow \pi(\hat{c})$
    Save state to metadata: $\mathsf{save}(c, \theta, \hat{c}, |\pi'|)$
    Compute loss $L'$
    **if** $L' < L$ **then**
        $(c, \pi, L) \leftarrow (\hat{c}, \pi', L')$ //update state
    **else**
        **break**
    **end if**
**end for**
**return** $c$ //output final centroids as model

---

Although it is rare, it is possible for a Lloyd's iteration to result in a degenerate (empty) cluster. In this scenario, we have two reasonable options. All of the theoretical guarantees are remain valid under both of the following options. The first option is to re-initialize a new cluster via a $k$-means++ seeding. Since the number of clusters $k$ and iterations $T$ are constant, this does not impact any of the asymptotic deletion efficiency results. The second option is to simply leave a degenerate partition. This does not impact the upper bound on expected statistical performance which is derived only as a function of the $k$-means++ initialization. For most datasets, this issue hardly matters in practice, since Lloyd's iterations do not usually produce degenerate clusters (even the presence of quantization noise). In our implementation, we have chosen to re-initialize degenerate clusters, and are careful to account for this in our metadata, since such a re-initialization could trigger the need to retrain at deletion time if the re-initialized point is part of the deletion stream.

We present the pseudo-code for the deletion operation (Algo. 5), and then elaborate on the quantization scheme and balance correction.

---

**Algorithm 5** Deletion Op for Q-$k$-means

---

**Input:** data matrix $D \in \mathbf{R}^{n \times d}$, target deletion index $i$, training metadata
Obtain target deletion point $p \leftarrow D_i$
Retrieve initial centroids from metadata: $\mathsf{load}(c_0)$
**if** $p \in c_0$ **then**
    // Selected initial point.
    **return** Q-k-means$(D_{-i})$ // Need to retrain from scratch.
**else**
    **for** $\tau = 1$ **to** $T$ **do**
        Retrieve state for iteration $\tau$: $\mathsf{load}(c,\theta,\hat{c},|\pi|)$
        Cluster assignment of $p$: $\kappa \leftarrow \mathbf{argmin}_k |p - c_k|$
        Perturbed centroid: $c'_\kappa \leftarrow c_\kappa - p/|\pi(c_\kappa)|$
        Apply $\gamma$-correction to $c'_\kappa$ if necessary
        Quantize perturbed centroid: $\hat{c}'_\kappa \leftarrow Q(c'_\kappa; \theta)$
        **if** $\hat{c}'_\kappa \neq \hat{c}_\kappa$ **then**
            // Centroid perturbed $\rightarrow$ unstable quantization
            **return** Q-k-means$(D_{-i})$ // Need to retrain from scratch.
        **end if**
        Update metadata with perturbed state: $\mathsf{save}(c',\theta,\hat{c}',|\pi'|)$
    **end for**
**end if**
Update $D \leftarrow D_{-i}$
**return** $\hat{c}$ //Successfully verified centroid stability

---

We proceed elaborate on the details of the balance correction and quantization steps

### B.1.1 $\gamma$-Balanced Clusters

**Definition B.1.** $\gamma$-Balanced

Given a partition $\pi$, we say it us $\gamma$-*balanced* if $|\pi_\kappa| \geq \frac{\gamma n}{k}$ for all partitions $\kappa$. The partition is $\gamma$-imbalanced if it is not $\gamma$-balanced.

In Q-$k$-means, imbalanced partitions can lead to unstable quantized centroids. Hence, it is preferable to avoid such partitions. As can be seen in the pseudo-code, we add mass to small clusters to correct for $\gamma$-unbalance. At each iteration we apply the following formula on all clusters such that $|\pi_\kappa| \geq \frac{\gamma n}{k}$: $c_\kappa \leftarrow |\pi(c_\kappa)|c_\kappa + (\frac{\gamma n}{k} - |\pi(c_\kappa)|)c'_\kappa$ where $c'$ denotes the centroids from the previous iteration.

In prose, for small clusters, current centroids are averaged with the centroids from the previous iteration to increase stability.

For use in practice, a choice of $\gamma$ must be made. If no class balance information is known, then, based on our observations, setting $\gamma = 0.2$ is a solid heuristic for all but severely imbalanced datasets, in which case it is likely that DC-$k$-means would be preferable to Q-$k$-means.

### B.1.2 Quantizing with an $\epsilon$-Lattice

We detail the quantization scheme used. A quantization maps analog values to a discrete set of points. In our scheme, we uniformly cover $\mathbf{R}^d$ with an $\epsilon$-lattice, and round analog values to the nearest vertex

on the lattice. It is also important to add an independent, uniform random phase shift to each dimension of lattice, effectively de-biasing the quantization.

We proceed to formally define our quantization $Q_{(\epsilon,\theta)}$. $Q_{(\epsilon,\theta)}$ is parameterized by a phase shift $\theta \in [-\frac{1}{2}, \frac{1}{2}]^d$ and an granularity parameter $\epsilon > 0$. For brevity, we omit the explicit dependence on phase and granularity when it is clear from context. For a given $(\epsilon, \theta)$:

$$a(x) = \mathbf{argmin}_{j \in \mathbf{Z}^d} \{||x - \epsilon(\theta + j)||_2\}$$
$$Q(x) = \epsilon(\theta + a(x))$$

We set $\{\theta\}_0^t$ with an iid random sequence such that $\theta_\tau \sim \mathbf{Unif}[-\frac{1}{2}, \frac{1}{2}]^d$.

## B.2 Divide-and-Conquer $k$-Means

We present pseudo-code for the deletion operation of divide-and-conquer $k$-means. The pseudo-code for the training algorithm may be found in the main body. The deletion update is conceptually simple. Since a deleted datapoint only belong to one leaf's dataset, we only need recompute the sub-problems on the path from said leaf to the root.

---

**Algorithm 6** Deletion Op for DC-$k$-means

---

**Input:** data matrix $D \in \mathbf{R}^{n \times d}$, target deletion index $i$, model metadata $M$
Obtain target deletion point $p \leftarrow D_i$
node $\leftarrow$ leaf node assignment of $p$
node.dataset $\leftarrow$ node.dataset $\setminus p$
**while** node is not root **do**
    node.parent.data $\leftarrow$ node.parent.data $\setminus$ node.centroids
    node.centroids $\leftarrow$ k-means++(node.data,$k$,$T$)
    node.parent.dataset.add(node.centroids)
    node $\leftarrow$ node.parent
**end while**
node.centroids $\leftarrow$ k-means++(node.dataset,$k$,$T$)
Update $D \leftarrow D_{-i}$
**return** node.centroids

---

## B.3 Initialization for $k$-Means

For both of our algorithms, we make use of the $k$-means++ initialization scheme [5]. This initialization is commonplace, and is the standard initialization in many scientific libraries, such as Sklearn [66]. In order to provide a more self-contained presentation, we provide some pseudo-code for the $k$-means++ initialization.

---

**Algorithm 7** Initialization by $k$-means++

---

**Input:** data matrix $D \in \mathbf{R}^{n \times d}$, number of clusters $k$
$i \leftarrow \mathbf{Uni}\{1,...,n\}$
$I \leftarrow \{D_i\}$.
$\mathbf{u} \leftarrow 0^n \in \mathbf{R}^n$.
**for** $1 < l \leq k$ **do**
    $u_j = \min_{\eta \in I} ||\eta - D_j||^2$ for all $1 \leq j \leq n$
    $Z = \sum_{j=1}^n u_j$
    Sample $i \sim \frac{1}{Z}\mathbf{u}$
    $I \leftarrow I \cup \{D_i\}$
**end for**
**return** $I$

---

# C   Mathematical Details

Here, we provide proofs for the claims in the main body. We follow notation introduced in the main body and Appendix A. As a notational shorthand, we will let denote $A(D_{-i})$ by $A_{-i}$ and $R(D,A(D),D_{-i})$ as $R$ when there is only one dataset in the context. Also, when it is unambiguous, we will use $A$ to denote the *specific* learning algorithm in question, and $R$ to denote its corresponding deletion operation.

## C.1 Proof of Theorem 4.1

Refer to the main body for the statement of Theorem 4.1. Here is an abridged version:

**Theorem.** *Q-k-means supports $m$ deletions in expected time $O(m^2 d^{5/2}/\epsilon)$.*

Note that we assume the dataset is scaled onto the unit hypercube. Otherwise, the theorem still holds with an assumed constant factor radial bound. We prove the theorem in three successive steps, given by Lemma C.1 through Lemma C.3.

**Lemma C.1.** *Define $C = [-\frac{\epsilon}{2}, \frac{\epsilon}{2}]^d$ for some $\epsilon > 0$. $C$ is the hypercube in Euclidean $d$-space of side length $\epsilon$ centered at the origin. Let $C' = [\frac{-\epsilon+\epsilon'}{2}, \frac{\epsilon-\epsilon'}{2}]^d$ for some $\epsilon' < \epsilon$. Let $X$ be a uniform random variable with support $C$. Then, $\mathbf{Pr}[X \in C/C'] \leq \frac{2d\epsilon'}{\epsilon}$.*

*Proof. (Lemma C.1)* If $X \in C/C'$, then there exists some $i \in \{1,...,d\}$ such that $X_i \in [-\frac{\epsilon}{2}, \frac{-\epsilon+\epsilon'}{2}] \cup [\frac{\epsilon-\epsilon'}{2}, \frac{\epsilon}{2}]$. Marginally, $\mathbf{Pr}[X_i \in [-\frac{\epsilon}{2}, \frac{-\epsilon+\epsilon'}{2}] \cup [\frac{\epsilon-\epsilon'}{2}, \frac{\epsilon}{2}]] = 2\epsilon'/\epsilon$. Taking a union bound over the $d$ dimensions obtains the bound. $\square$

We make use of Lemma C.1 in proving the following lemma. First, recall the definition of our quantization scheme $Q$ from Section 3:
$$Q(x) = \epsilon(\theta + \mathbf{argmin}_{j \in \mathbf{Z}^d}\{||x - \epsilon(\theta+j)||_2\}).$$
We take $\theta \sim \mathbf{Uni}[-\frac{1}{2}, \frac{1}{2}]^d$, implying a distribution for $Q$.

**Lemma C.2.** *Let $Q$ be a uniform quantization $\epsilon$-lattice over $\mathbf{R}^d$ with uniform phase shift $\theta$. Let $Q(\cdot)$ denote the quantization mapping over $\mathbf{R}^d$ and let $Q[\cdot]$ denote the quantization image for subsets of $\mathbf{R}^d$. Let $X \in \mathbf{R}^d$. Then $\mathbf{Pr}[\{Q(X)\} \neq Q[B_{\epsilon'}(X)]] < \frac{2d\epsilon'}{\epsilon}$, where $B_{\epsilon'}(X)$ is the $\epsilon'$-ball about $X$ under Euclidean norm.*

*Proof. (Lemma C.2)* Due to invariance of measure under translation, we may apply a coordinate transformation by translation $Q(X)$ to the origin of $\mathbf{R}^d$. Under this coordinate transform, $X \sim \mathbf{Uni}[-\frac{\epsilon}{2}, \frac{\epsilon}{2}]^d$. Further, note that $\mathbf{Pr}[B_{\epsilon'}(X) \subset [-\frac{\epsilon}{2}, \frac{\epsilon}{2}]^d]$ is precisely equivalent to $\mathbf{Pr}[X \in [\frac{-\epsilon+\epsilon'}{2}, \frac{\epsilon-\epsilon'}{2}]^d]$. Because $X$ is uniform, applying Lemma C.1 as an upper bound completes the proof. $\square$

With Lemma C.2 in hand, we proceed to state and prove Lemma C.3.

**Lemma C.3.** *Let $D$ be an dataset on $[0,1]^d$ of size $n$. Let $\overline{\mathbf{c}}(D)$ be the centroids computed by Q-k-means with initialization $I$ and parameters $T$, $k$, $\epsilon$, and $\gamma$. Then, with probability greater than $1 - \frac{2mTkd^{3/2}}{\epsilon\gamma n}$, it holds that $\overline{\mathbf{c}}(D) = \overline{\mathbf{c}}(D_{-\Delta})$ for any $\Delta \subset D$ with $|\Delta| \leq m$ and $\Delta \cap I = \emptyset$, where probability is with respect to the randomness in the quantization phase.*

*Proof. (Lemma C.3)* We analyze two instances of Q-$k$-means algorithm operating with the same initial centroids and the same sequence of iid quantization maps $\{Q_\tau\}_1^t$. One instance runs on input dataset $D$ and the other runs on input dataset $D_{-\Delta}$. This is the only difference between the two instances.

Let $\mathbf{c}_I^{(\tau,\kappa)}(\delta)$ denote the $\kappa$-th analog (i.e. non-quantized) centroid at the $\tau$-th iteration of Q-$k$-means on some input dataset $\delta$ with initialization $I$. By construction, for any datasets $\delta, \delta'$, we have that $\overline{\mathbf{c}}_I(\delta) = \overline{\mathbf{c}}_I(\delta')$ if $Q_\tau(\mathbf{c}_I^{(\tau,\kappa)}(\delta)) = Q_\tau(\mathbf{c}_I^{(\tau,\kappa)}(\delta'))$ for all $\tau \in \{1,...,t\}$ and all $\kappa \in \{1,...,k\}$.

Fix any particular $\tau$ and $\kappa$. We can bound $||\mathbf{c}_I^{(\tau,\kappa)}(D) - \mathbf{c}_I^{(\tau,\kappa)}(D_{-\Delta})||_2$ as follows. Note that $\mathbf{c}_I^{(\tau,\kappa)}(D) = \frac{1}{n}\sum_{i=1}^{n} D_i \mathbf{1}(D_i \in \pi_\kappa)$ where $\mathbf{1}(\cdot)$ denotes the indicator function. Furthermore, $\mathbf{c}_I^{(\tau,\kappa)}(D_{-\Delta}) = \frac{1}{n}\sum_{i=1}^{n} D_i \mathbf{1}(D_i \in \pi_\kappa)\mathbf{1}(D_i \notin \Delta)$. Assume that $|\pi_\kappa| \geq \gamma n/k$. Because $|\Delta| \leq m$ and $||D_i||_2 \leq \sqrt{d}$, these sums can differ by at most $\frac{mk\sqrt{d}}{\gamma n}$. On the other hand, assume that $|\pi_\kappa| < \gamma n/k$. In this case, the $\gamma$-correction still ensures that the sums differ by at most $\frac{mk\sqrt{d}}{\gamma n}$. This bounds $||\mathbf{c}_I^{(\tau,\kappa)}(D) - \mathbf{c}_I^{(\tau,\kappa)}(D - \Delta)||_2 \leq \frac{mk\sqrt{d}}{\gamma n}$.

To complete the proof, apply Lemma C.2 setting $\epsilon' = \frac{mk\sqrt{d}}{\gamma n}$. Taking a union bound over $\tau \in \{1,...,t\}$ and $\kappa \in \{1,...,k\}$ yields the desired result.

$\square$

We are now ready to complete the theorem. We briefly sketch and summarize the argument before presenting the proof. Recall the deletion algorithm for Q-$k$-means (Appendix B). Using the runtime memo, we verify that the deletion of a point does not change what would have been the algorithm's output. If it would have changed the output, then we retrain the entire algorithm from scratch. Thus, we take a weighted average of the computation expense in these two scenarios. Recall that retraining from scratch takes time $O(nkTd)$ and verifying the memo at deletion time takes time $O(kTd)$. Finally, note that we must sequentially process a sequence of $m$ deletions, with a valid model output after each request. We are mainly interested in the scaling with respect to $m$, $\epsilon$ and $n$, treating other factors as non-asymptotic constants in our analysis. We now proceed with the proof.

**Theorem.** *Q-$k$-means supports $m$ deletions in expected time $O(m^2 d^{5/2}/\epsilon)$.*

*Proof. (Correctness)*
In order for $R$ to be a valid deletion, we require that $R =_d A_{-i}$ for any $i$. In this setting, we identify models with the output centroids: $A(D) = \bar{c}_I(D)$. Consider the sources of randomness: the iid sequence of random phases and the $k$-means++ initializations.

Let $I(\cdot)$ be a set-valued random function computing the $k$-means++ initializations over a given dataset. $E$ denote the event that $D_i \notin I(D)$. Then, from the construction of $k$-means++, we have that for all $j \neq i$, $\mathbf{Pr}[D_j \in I(D_{-i})] = \mathbf{Pr}[D_j \in I(D)|E]$. Thus, $I(D)$ and $I(D_{-i})$ are equal in distribution conditioned on $i$ not being an initial centroid. Note that this is evident from the construction of $k$-means++ (see algorithm 7).

Let $\theta$ denote the iid sequence of random phases for $A$ and let $\theta_{-i}$ denote the iid sequence of random phases for $A_{-i}$. Within event $E$, we define a set of events $E'(\hat{\theta})$, parameterized by $\hat{\theta}$, as the event that output centroids are stable under deletion conditioned on a given sequence of phases $\hat{\theta}$:

$$E = \{i \notin I(D)\}, E'(\hat{\theta}) = \{A|\{\theta = \hat{\theta}\} = A_{-i}|\{\theta_{-i} = \hat{\theta}\}\} \cap E$$

By construction of $R$, we have $R = A\mathbf{1}(E'(\theta)) + A_{-i}\mathbf{1}(\overline{E'(\theta)})$ where event $E'(\theta)$ is verified given the training time memo. To conclude, let $S$ be any Borel set:

$\mathbf{Pr}[R \in S] = \mathbf{Pr}[E'(\theta)]\mathbf{Pr}[R \in S|E'(\theta)] + (1 - \mathbf{Pr}[E'(\theta)])\mathbf{Pr}[R \in S|\overline{E'(\theta)}]$ by law of total probability.

$= \mathbf{Pr}[E'(\theta)]\mathbf{Pr}[A \in S|E'(\theta)] + (1 - \mathbf{Pr}[E'(\theta)])\mathbf{Pr}[A_{-i} \in S]$ by construction of $R$

$= \mathbf{Pr}[E'(\theta)]\mathbf{Pr}[A_{-i} \in S|\theta_{-i} = \theta] + (1 - \mathbf{Pr}[E'(\theta)])\mathbf{Pr}[A_{-i} \in S]$ by definition of $E'$

$= \mathbf{Pr}[E'(\theta)]\mathbf{Pr}[A_{-i} \in S] + (1 - \mathbf{Pr}[E'(\theta)])\mathbf{Pr}[A_{-i} \in S] = \mathbf{Pr}[A_{-i} \in S]$ by $\theta =_d \theta_{-i}$.

$\square$

*Proof. (Runtime)*

Let $\mathcal{T}$ be the total runtime of $R$ after training $A$ once and then satisfying $m$ deletion requests with $R$. Let $\Delta = \{i_1, i_2, ..., i_m\}$ denote the deletion sequence, with each deletion sampled uniformly without replacement from $D$.

Let $\Psi$ be the event that the centroids are stable for all $m$ deletions. Using Lemma C.3 to bound the event complement probability $\mathbf{Pr}(\overline{\Psi})$:

$$\mathbf{E}\mathcal{T} \leq \mathbf{E}[\mathcal{T}|\Psi] + \mathbf{Pr}[\overline{\Psi}]\mathbf{E}[\mathcal{T}|\overline{\Psi}] = O(mkTd) + O(\epsilon^{-1}m^2T^2k^3d^{2.5}) = O(m^2 d^{2.5}/\epsilon).$$

In $\Psi$ the centroids are stable, and verifying in $\Psi$ takes time $O(mktd)$ in total. In $\overline{\Psi}$ we coarsely upper bounded $\mathcal{T}$ by assuming we re-train to satisfy each deletion. $\square$

### C.2 Proofs of Corollaries and Propositions

We present the proofs of the corollaries in the main body. We are primarily interested in the asymptotic effects of $n$, $m$, $\epsilon$, and $w$. We treat other variables as constants. For the purposes of online analysis, we let $\epsilon = \Theta(n^{-\beta})$ for some $\beta \in (0,1)$ and $w = \Theta(n^\rho)$ for some $\rho \in (0,1)$

### C.2.1 Proof of Corollory 4.1.1

We state the following Theorem of Arthur and Vassilvitskii concerning $k$-means++ initializations [5]:

**Theorem C.4.** *(Arthur and Vassilivitskii)*

*Let $\mathcal{L}^*$ be the optimal loss for a $k$-means clustering problem instance. Then $k$-means++ achieves expected loss $\mathbf{E}\mathcal{L}^{++} \leq (8\ln k + 16)\mathcal{L}^*$*

We re-state corollary 4.1.1:

**Corollary.** *Let $\mathcal{L}$ be a random variable denoting the loss of $Q$-$k$-means on a particular problem instance of size $n$. Then $\mathbf{E}\mathcal{L} \leq (8\ln k + 16)\mathcal{L}^* + \epsilon\sqrt{nd(8\ln k + 16)\mathcal{L}^*} + \frac{1}{4}nd\epsilon^2$.*

*Proof.* Let $c$ be the initialization produced $k$-means++. Let $\mathcal{L}^{++} = \sum_{i=1}^{n}||c(i) - x_i||_2^2$ where $c(i)$ is the centroid closest to the $i$-th datapoint. Let $||c(i) - x_i||_2^2 = \sum_{j=1}^{d}\delta_{ij}^2$, with $\delta_{ij}$ denoting the scalar distance between $x_i$ and $c(i)$ in the $j$-th dimension. Then we may upper bound $\mathcal{L} \leq \sum_{i=1}^{n}\sum_{j=1}^{d}(\delta_{ij}^2 + \frac{1}{4}\epsilon^2 + \delta_{ij}\epsilon)$ by adding a worst-case $\frac{\epsilon}{2}$ quantization penalty in each dimension. This sum reduces to:

$\mathcal{L} \leq \sum_{i}^{n}\sum_{j}^{d}\delta_{ij}^2 + \sum_{i}^{n}\sum_{j}^{d}\frac{1}{4}\epsilon^2 + \sum_{i}^{n}\sum_{j}^{d}\delta_{ij}\epsilon = \mathcal{L}^{++} + \frac{1}{4}nd\epsilon^2 + \epsilon\sqrt{nd\mathcal{L}^{++}}$. The third term comes from the fact that $\sqrt{nd\mathcal{L}^{++}} \geq \sum_{i}^{n}\sum_{j}^{d}\delta_{ij} \geq \sqrt{\mathcal{L}^{++}}$ if $\sum_{i}^{n}\sum_{j}^{d}\delta_{ij}^2 = \mathcal{L}^{++}$ and $\delta_{ij} > 0$ (to see this, treat it as a constrained optimization over the $\delta_{ij}$). Thus:

$$\mathbf{E}\mathcal{L} \leq \mathbf{E}\mathcal{L}^{++} + \frac{1}{4}nd\epsilon^2 + \epsilon\sqrt{nd}\mathbf{E}\sqrt{\mathcal{L}^{++}}$$

Using Jensen's inequality [26] yields $\mathbf{E}\sqrt{\mathcal{L}} \leq \sqrt{\mathbf{E}\mathcal{L}}$:

$$\mathbf{E}\mathcal{L} \leq \mathbf{E}\mathcal{L}^{++} + \frac{1}{4}nd\epsilon^2 + \epsilon\sqrt{nd}\sqrt{\mathbf{E}\mathcal{L}^{++}}$$

To complete the proof, apply Theorem C.4:

$$\mathbf{E}\mathcal{L} \leq C\mathcal{L}^* + \frac{1}{4}nd\epsilon^2 + \epsilon\sqrt{ndC\mathcal{L}^*}$$

where $C = 8\ln k + 16$. $\square$

### C.2.2 Proof of Proposition 4.2

**Proposition.** *Let $D$ be an dataset on $\mathbf{R}^d$ of size $n$. Fix parameters $T$ and $k$ for DC-$k$-means. Let $w = \Theta(n^\rho)$ and $\rho \in (0,1)$ Then, with a depth-1, $w$-ary divide-and-conquer tree, DC-$k$-means supports $m$ deletions in time $O(mn^{\max(\rho, 1-\rho)}d)$ in expectation with probability over the randomness in dataset partitioning.*

*Proof. (Correctness)* We require that $R(D, A(D), i) =_d A(D_{-i})$. Since each datapoint is assigned to a leaf independently, the removal of a datapoint does not change the distribution of the remaining datapoints to leaves. However, one must be careful when it comes to the number of leaves, which cannot change due to a deletion. This is problematic if the number of leaves is $\lceil n^\rho \rceil$ (or another similar quantity based on $n$).

The simplest way to address this (without any impact on asymptotic rates) is to round the number of leaves to the nearest power of 2. This works because the intended number of leaves will only be off from $n^\rho$ by at most a factor of 2. In the rare event this rounding changes due to a deletion, we will have to default to retraining from scratch, but, asymptotically in the fractional power regime, this can only happen a constant number of times which does not affect an amortized or average-case time complexity analysis. $\square$

We proceed to prove the runtime analysis.

*Proof. (Runtime)*

Let $\mathcal{T}$ be the total runtime of $R$ after training $A$ once and then satisfying $m$ deletion requests with $R$. Let $\Delta = \{i_1, i_2, ..., i_m\}$ denote the deletion sequence, with each deletion sampled uniformly without replacement from $D$.

Let $S$ be the uniform distribution over $n^\rho$ elements and let $\hat{S}$ be the empirical distribution of $n$ independent samples of $S$. The fraction of datapoints assigned to the $i$-th leaf is then modeled by $\hat{S}_i$. We treat $\hat{S}$ as probability vector. Let random variable $J = n\hat{S}_i$ with probability $\hat{S}_i$. Thus, $J$ models the distribution over sub-problem sizes for a randomly selected datapoint. Direct calculation yields the following upper bound on runtime:

$\mathcal{T} \leq m(O(kTdJ) + O(n^\rho k^2 Td))$ where the first term is due to the total deletion time at the leaves, the second term is due to the total deletion time at the root, and the $m$ factor is due to the number of deletions.

Hence, we have $\mathbf{E}(\mathcal{T}) \leq O(mkTd)\mathbf{E}(J) + O(mn^\rho k^2 Td)$, with $\mathbf{E}(J)$ representing the quantity of interest. Computing $\mathbf{E}(J)$ is simple using the second moments of the Binomial distribution , denoted by $\mathcal{B}$:

$$\mathbf{E}(J) = \mathbf{E}(\mathbf{E}(J|\hat{S})) = \sum_{i=1}^{n^\rho} n\mathbf{E}(\hat{S}_i^2)$$

Noting that $\hat{S}_i \sim \frac{1}{n}\mathcal{B}(n, n^{-\rho})$ and $\mathbf{E}((\mathcal{B}(n,p)^2) = n(n-1)p + np$ [48] yields:

$$= n^{\rho-1}\mathbf{E}((\mathcal{B}(n,n^{-\rho})^2) = O(n^{1-\rho})$$

This yields the final bound: $\mathbf{E}(\mathcal{T}) \leq O(n^{1-\rho}mkTd) + O(n^\rho mk^2 Td) = O(m\mathbf{max}\{n^{1-\rho}, n^\rho\}d)$

$\square$

### C.2.3   Proof of Corollary 4.2.1

**Corollary.** *With $\epsilon = \Theta(n^{-\beta})$ for $0 < \beta < 1$, Q-k-means algorithm is deletion efficient in expectation if $\alpha \leq \frac{1-\beta}{2}$*

*Proof.* We are interested in the asymptotic scaling of $n$, $m$, and $\epsilon$, and treat other factors as cosntants. We begin with the expected deletion time from Theorem 4.1, given by $O(m^2 d^{5/2}\epsilon^{-1})$. Recall we are using rates $\epsilon = \Theta(n^{-\beta})$ and $m = \Theta(n^\alpha)$. Applying the rates, adding in the training time, and amortizing yields $O(n^{1-\alpha} + n^{\alpha+\beta})$. Thus, deletion efficiency follows if $1 - \alpha > \alpha + \beta$. Rearranging terms completes the calculation. $\square$

### C.2.4   Proof of Corollary 4.2.2

**Corollary.** *With $w = \Theta(n^\rho)$ and a depth-1 $w$-ary divide-and-conquer tree, DC-k-means is deletion efficient in expectation if $\alpha \leq 1 - \mathbf{max}(\rho, 1-\rho)$*

*Proof.* We are interested in the asymptotic scaling of $n$, $m$, and $w$, and treat other factors as constants. Recall we are using rates $w = \Theta(n^\rho)$ and $m = \Theta(n^\alpha)$ By Proposition 4.2, the runtime of each deletion is upper bounded by $O(n^{\mathbf{max}(\rho, 1-\rho)})$ and the training time is $O(n)$. Amortizing and comparing the terms yields the desired inequaltiy. Deletion efficiency follows if $\mathbf{max}\{\rho, 1-\rho\} \leq 1 - \alpha$. Rearranging terms completes the calculation. $\square$

## D   Implementation and Experimental Details

We elaborate on implementation details, the experimental protocol used in the main body, and present some supplementary experiments that inform our understanding of the proposed techniques.

### D.1   Experimental Protocol

We run a $k$-means baseline (i.e. a $k$-means++ seeding followed by Lloyd's algorithm), Q-$k$-means, and DC-$k$-means on 6 datasets in the simulated online deletion setting. As a proxy for deletion efficiency, we report the wall-clock time of the program execution on a single-core of an Intel Xeon E5-2640v4 (2.4GHz) machine. We are careful to only clock the time used by the algorithm and pause the clock when executing test-bench infrastructure operations. We do not account for random OS-level

interruptions such as context switches, but we are careful to allocate at most one job per core and we maintain high CPU-utilization throughout.

For each of the three methods and six datasets, we run five replicates of each benchmark to obtain standard deviation estimates. To initialize the benchmark, each algorithm trains on the complete dataset, which is timed by wall-clock. We then evaluate the loss and clustering performance of the centroids (untimed). Then, each model must sequentially satisfy a sequence of 1,000 uniformly random (without replacement) deletion requests. The time it takes to satisfy each request is also timed and added to the training time to compute a total computation time. The total computation time of the benchmark is then amortized by dividing by 1,000 (the number of deletion requests). This produces a final amortized wall-clock time. For the $k$-means baseline, we satisfy deletion via naive re-training. For Q-$k$-means and DC-$k$-means we use the respective deletion operations. As part of our benchmark, we also evaluate the statistical performance of each method after deletions 1,10,100, and 1,000. Since we are deleting less than 10% of any of our datasets, the statistical performance metrics do not change significantly throughout the benchmark and neither do the training times (when done from scratch). However, a deletion operation running in time significantly less than it takes to train from scratch should greatly reduce the total runtime of the benchmark. Ideally, this can be achieved without sacrificing too much cluster quality, as we show in our results (Section 5).

### D.1.1 Implementation Framework

We are interested in *fundamental* deletion efficiency, however, empirical runtimes will always be largely implementation specific. In order to minimize the implementation dependence of our results, we control by implementing an in-house version of Lloyd's iterations which is used as the primary optimization sub-routine in all three methods. Our solver is based on the Numpy Python library [82]. Thus, Q-$k$-means and DC-$k$-means use the same sub-routine for computing partitions and centroids as does the $k$-means baseline. Our implementation for all three algorithms can be found at `https://github.com/tginart/deletion-efficient-kmeans`.

### D.1.2 Heuristic Parameter Selection

Hyperparameter tuning poses an issue for deletion efficiency. In order to be compliant to the strictest notions of deletion, we propose the following *heuristics* to select the quantization granularity parameter $\epsilon$ and the number of leaves $w$ for Q-$k$-means and DC-$k$-means, respectively. Recall that we always set iterations to 10 for both methods.

*Heurstic Parameter Selection for Q-$k$-means*. Granularity $\epsilon$ tunes the centroid stability versus the quantization noise. Intuitively, when the number of datapoints in a cluster is high compared to the dimension, we need lower quantization noise to stabilize the centroids. A good rule-of-thumb is to use $\epsilon = 2^{\lfloor -\log_{10}(\frac{n}{kd^{3/2}})-3 \rfloor}$, which yields an integer power of 2. The heuristic can be conceptualized as capturing the effective cluster mass per dimension of a dataset. We use an exponent of $1.5$ for $d$, which scales like the stability probability (see Lemmas C.1 - C.3). The balance correction parameter $\gamma$ is always set to 0.2, which should work well for all but the most imbalanced of datasets.

*Heurstic Parameter Selection for DC-$k$-means*. Tree width $w$ tunes the sub-problem size versus the number of sub-problems. Intuitively, it is usually better to have fewer larger sub-problems than many smaller ones. A good rule-of-thumb is to set $w$ to $n^{0.3}$, rounded to the nearest power of two.

### D.1.3 Clustering Performance Metrics

We evaluate our cluster quality using the silhouette coefficient and normalized mutual information, as mentioned in the main body. To do this evaluation, we used the routines provided in the Scikit-Learn Python library [65]. Because computing the silhouette is expensive, for each instance we randomly sub-sample 10,000 datapoints to compute the score.

### D.1.4 Scaling

We note that all datasets except MNIST undergo a minmax scaling in order to map them into the unit hypercube (MNIST is already a scaled greyscale image). In our main body, we treat this as a one-time scaling inherit to the dataset itself. In practice, the scaling of a dataset can change due to deletions. However, this is a minor concern (at least for minmax scaling) as only a small number of extremal datapoints affect the scale. Retraining from scratch when these points come up as a deletion request does not impact asymptotic runtime, and has a negligible impact on empirical runtime. Furthermore, we point out that scaling is not necessary for our methods to work. In fact, in datasets where the notion

of distance remains coherent across dimensions, one should generally refrain from scaling. Our theory holds equally well in the case of non-scaled data, albeit with an additional constant scaling factor such as a radial bound.

## D.2 Datasets

- `Celltypes` [42] consists of $12,009$ single cell RNA sequences from a mixture of $4$ cell types: microglial cells, endothelial cells, fibroblasts, and mesenchymal stem cells. The data was retrieved from the Mouse Cell Atlas and consists of 10 feature dimensions, reduced from an original $23,433$ dimensions using principal component analysis. Such dimensionality reduction procedures are a common practice in computational biology.
- `Postures` [35, 34] consists of $74,975$ motion capture recordings of users performing $5$ different hand postures with unlabeled markers attached to a left-handed glove.
- `Covtype` [12] consists of $15,120$ samples of $52$ cartographic variables such as elevation and hillshade shade at various times of day for 7 forest cover types.
- `Botnet` [56] contains statistics summarizing the traffic between different IP addresses for a commercial IoT device (Danmini Doorbell). We aim to distinguish between benign traffic data ($49,548$ instances) and 11 classes of malicious traffic data from botnet attacks, for a total of $1,018,298$ instances.
- `MNIST` [51] consists of $60,000$ images of isolated, normalized, handwritten digits. The task is to classify each $28 \times 28$ image into one of the ten classes.
- `Gaussian` consists of 5 clusters, each generated from 25-variate Gaussian distribution centered at randomly chosen locations in the unit hypercube. $20,000$ samples are taken from each of the 5 clusters, for a total of $100,000$ samples. Each Gaussian cluster is spherical with variance of $0.8$.

## D.3 Supplementary Experiments

We include three supplementary experiments. Our first is specific to Q-$k$-means (See Appendix D.3.1), and involves the stability of the quantized centroids against the deletion stream. In our second experiment we explore how the choices of key parameters ($\epsilon$ and $w$) in our proposed algorithms contribute to the statistical performance of the clustering. In our third experiment, we explore how said choices contribute to the deletion efficiency in the online setting.

### D.3.1 Re-training During Deletion Stream for Q-$k$-means

In this experiment, we explore the stability of the quantized centroids throughout the deletion stream. This is important to understand since it is a fundamental behavior of the Q-$k$-means, and is not an implementation or hardware specific as a quantity like wall-clock time is. We plot, as a function of deletion request, the average number of times that Q-$k$-means was forced to re-train from scratch to satsify a deletion request.

Figure 2: Average retrain occurrences during deletion stream for Q-$k$-means

As we can see in Fig. 2, when the effective dimensionality is higher (relative to sample size), like in the case of `MNIST`, our retraining looks like somewhat of a constant slope across the deletions, indicating that the quantization is unable to stabilize the centroids for an extended number of deletion requests.

### D.3.2    Effects of Quantization Granularity and Tree Width on Optimization Loss

Although the viability of hyperparameter tuning in the context of deletion efficient learning is dubious, from a pedagogical point of view, it is still interesting to sweep the main parameters (quantization granularity $\epsilon$ and tree width $w$) for the two proposed methods. In this experiment, we compare the $k$-means optimization loss for a range of $\epsilon$ and $w$. As in the main body, we normalize the $k$-means objective loss to the baseline and restrict ourselves to depth-1 trees.

Figure 3: Loss Ratio vs. $\epsilon$ for Q-$k$-means on 6 datasets

Figure 4: Loss Ratio vs. $w$ for DC-$k$-means on 6 datasets

In Fig. 3, Q-$k$-means performance rapidly deteriorates as $\epsilon \to 1$. This is fairly expected given our theoretical analysis, and is also consistent across the six datasets.

On the other hand, in Fig. 4, we see that the relationship between $w$ and loss is far weaker. The general trend among the datasets is that performance decreases as width increases, but this is not always monotonically the case. As was mentioned in the main body, it is difficult to analyze relationship between loss and $w$ theoretically, and, for some datasets, it seems variance amongst different random seeds can dominate the impact of $w$.

### D.3.3 Effects of Quantization Granularity and Tree Width on Deletion Efficiency

On the Covtype dataset, we plot the amortized runtimes on the deletion benchmark for a sweep of $\epsilon$ and $w$ for both Q-$k$-means and DC-$k$-means, respectively. As expected, the runtimes for Q-$k$-means monotonically increase as $\epsilon \to 0$. The runtimes for DC-$k$-means are minimized by some an optimal tree width at approximately 32-64 leaves.

Figure 5: Amortized runtime (seconds) for Q-$k$-means as a function of quantization granularity on Covtype

Figure 6: Amortized runtime (seconds) for DC-$k$-means as a function of tree width on Covtype

## E  Extended Discussion

We include an extended discussion for relevant and interesting ideas that are unable to fit in the main body.

### E.1  Deletion Efficiency vs. Statistical Performance

In relational databases, data is highly structured, making it easy to query and delete it. This is not the case for most machine learning models. Modern learning algorithms involve data processing that is highly complex, costly, and stochastic. This makes it difficult to efficiently quantify the effect of an individual datapoint on the entire model. Complex data processing may result in high-quality statistical learning performance, but results in models for which data deletion is inefficient, and, in the worst case, would require re-training from scratch. On the other hand, simple and structured data processing yields efficient data deletion operations (such as in relational databases) but may not boast as strong statistical performance. *This is the central difficulty and trade-off engineers would face in designing deletion efficient learning systems.*

Hence, we are primarily concerned with *deletion efficiency* and *statistical performance* (i.e. the performance of the model in its intended learning task). In principle, these quantities can both be measured theoretically or empirically. We believe that the amortized runtime in the proposed online deletion setting is a natural and meaningful way to measure deletion efficiency. For deletion time, theoretical analysis involves finding the amortized complexity in a particular asymptotic deletion regime. In the empirical setting, we can simulate sequences of online deletion requests from real datasets and measure the amortized deletion time on wall-clocks. For statistical performance, theoretical analysis can be difficult but might often take the shape of a generalization bound or an approximation ratio. In the empirical setting, we can take the actual optimization loss or label accuracy of the model on a real dataset.

### E.2  Overparametrization, Dimensionality Reduction and Quantization

One primary concern with quantization is that it performs poorly in the face of overparameterized models. In some situations, metric-preserving dimensionality reduction techniques [45, 28] could potentially be used.

### E.3 Hyperparameter Tuning

Hyperparameter tuning is an essential part of many machine learning pipelines. From the perspective of deletion efficient learning, hyperparameter tuning presents somewhat of a conundrum. Ultimately, in scenarios in which hyperparameter tuning does indeed fall under the scope of deletion, one of the wisest solutions may be to tune on a subset of data that is unlikely to be deleted in the near future, or to pick hyperparameters via good heuristics that do not depend on specific datapoints.