[Reviews · NeurIPS 2019]

Reviewer 1



The paper addresses "the right to be forgotten" and how an individual's data contribution to the trained model can be removed without having to retrain the model from scratch. I believe this work makes important contributions to the study of a real-world problem with significant practical and policy implications. The paper is generally well-written and well-structured, but I believe it could be made more accessible to the average reader of NeurIPS (See below). Authors have done a thorough literature review (in particular, I liked that they clearly distinguish their problem from privacy.) I very much appreciated the 4 principles they lay forward in the discussion section. I just wish they had made better references to those principles in the exposition of their algorithms.

Reviewer 2



*Originality* To my knowledge, the idea of designing deletion efficient machine learning algorithms is novel. The paper differentiates itself well from related work. In fact I was very impressed at how well it places itself among its lengthy list of references, which include all the key pieces to understand the work. *Quality* Overall the paper is quite high quality. The motivation for the problem is clear. The problem formulation is clear. A simple algorithm is chosen to illustrate the problem, and multiple solutions that illustrate different solution techniques are given. The empirical validation of the new algorithms proposed is thorough. I have only minor gripes that prevent me from offering my highest recommendation. The authors begin with a personal anecdote, and I thought the relevance of the problem to the authors would then lead to an example drawn from a practical situation involving a real trained system of the authors. Instead, the relatively toy example of k-means is used. I think the biggest weakness of this paper is that efficient data deletion is only really needed when retraining the algorithm is prohibitively slow, but k-means is actually very fast! The authors show a speedup from a matter or minutes to a matter of seconds, but is that difference actually practically meaningful? A real, practical context is needed to illustrate if it is. The argument of the paper would be much stronger if a deletion speedup was demonstrated that moved a system from being impractical to being practical. A second issue is conceptual. For some contexts, data deletion seems much harder than just removing a single user's data, such as when data is relational. For instance, suppose I want to delete my data from Facebook. The authors propose a way to remove information from data I have entered to Facebook's trained models, but what about information that other people have entered about me, such as photos of me where I am not tagged? Of course this issue is larger than just the machine learning problem of deleting data from learned models, but given the policy relevance of this work, I would warn the authors against claiming that a simple view of data deletion is feasible. *Clarity* The paper was clear and a pleasure to read. The only issue I had with the way the paper was organized was the choise of only studying k-means. *Significance* If it is the case and not just my own ignorance that deletion efficient algorithms is a novel area, I think this work has a high potential for impact. Given the policy-relevance of having efficient deletion operations available, I think this problem formulation could be as impactful as k-anonymity, for example, and I was excited as I was reviewing this paper for that reason. *Post-Rebuttal* I have now read in some detail through the appendix. Although I didn't work through every line, I at least find the arguments convincing. There are many small errors and other opportunities for improvement in the appendix, which I will list below. I recommend the authors do their own careful check of these materials as well. In Definition A.3 is it correct that adversarial deletion could also serve as a model for deletion of sets of correlated data, for example, which might otherwise be a concern in the average case analysis? The example that is currently given regarding an adversary always deleting a point in the first leaf is a bit confusing and poorly motivated. The differential privacy discussion in the appendix is quite interesting, and I wonder if at least another pointer can be added to the main text to emphasize that connection. Does the balancing operation affect the types of clusters that are found, even if not the loss so much? More discussion on that would be nice. Is Algorithm 4 guaranteed to converge with the loss reduction stopping condition? In the balancing operation, should the weighted average be normalized somehow, like maybe by n? It seems the weights right now are not between 0 and 1. Why is gamma = 0.2 a solid heuristic? Currently that comes out of nowhere in Section B.1.1. Also in regard to the same paragraph "In practice, a choice...", how do you know or find out if a dataset D is unbalanced? The Unif[-0.5,0.5] distribution on theta at the end of page 17 is different from the Unif[0,1] distribution given in Algorithm 4. Missing a parenthesis in the first paragraph of section C. The lemmas are all misnumbered in the text of section C. typo: "the what" You use both the names Q-Lloyd and Q-k-means. The reference to the lower bound on centroid stability in theorem 3.1 is confusing. Isn't that what you are proving there?

Reviewer 3



* Summary The authors study the problem of deletion efficient learning, i.e., given a dataset D and a model M, how to 'quickly' form the new model M' such that it is equal to a model trained on the dataset D' where some item has been removed? Here quickly means faster than the naive approach of retraining a model on the dataset D'. The paper contains a theoretical part and an empirical part. The theoretical part defines the problem and presenting two variants of the k-means algorithm that are deletion efficient. In the empirical part the authors investigate the performance of their algorithms and conclude that they are faster than the naive approach of retraining a model. * Originality The authors state the concept of deletion efficient learning is novel, and based on this it appears that the ideas presented in the paper are new. Also, the two algorithms appear to the best of my knowledge to be new variants of k-means. There is no separate 'related work' section in the paper, and I would encourage the authors to add a such, pointing to research that is (closely) related (such as the privacy-related matters mentioned in the paper). This would help the reader to better position the present work. * Quality I think that the technical quality of the paper is good. The problem being studied is clearly defined and the notation is also OK. The two algorithms are presented in sufficient details in the main paper. Proofs are not presented in the main manuscript and I did not check proofs in the appendix. I found the empirical evaluation OK, and I think it demonstrates the utility of the presented algorithms. In overall, I think that this is a well-written paper that is also well motivated. However, I think that the authors might want to "downplay" the data deletion efficient learning in machine learning as a general concept, since the focus in the main manuscript (I did not check Appendix E in detail) is in fact only on deletion efficient unsupervised learning, specifially k-means clustering. If the authors have some relevant points concerning supervised learning methods, I think it would be very important to bring these forward in the main paper. This is due to the fact that supervised learning algorithms are much used in various domains and I think that the impact of this paper would be greater if deletion efficient learning in the domain of supervised learning would be discussed in more detail in the manuscript. * Clarity The paper is well structured and the langauge is clear. The introduction is clearly written and provides a good motivation for the work. The experimental evaluation is also clearly presented. I found some minor typos: - Sec. 3.2, paragraph 1: "as and independent" --> "as an independent" - Sec. 4, Datasets: "we run our experiments five" --> "we run our experiments on five" - Table 4: MNSIT --> MNIST * Significance Summarising, I think that the ideas in this paper might be of interest to the machine learning community and can possibly spur new research. * Update after Author Feedback I have read the author feedback. I find it good that the authors will try to include some more discussion concerning supervised learning in the manuscript, which I think will make this paper even better.

[Author Response · NeurIPS 2019]

We would like to thank all of the reviewers for giving detailed and helpful feedback. We will incorporate the suggestions in the revised paper. We are glad that all the reviewers found the problem of data deletion interesting and important to study. In fact, just a few weeks ago, the Dashboard Act proposed in the U.S. Senate stipulates that large consumer companies need to enable users to delete their data, making this a timely and significant challenge to the ML community.

**Response to Reviewer 1:**    Thank you for your suggestions on making our exposition more accessible. In the revision, we will add a concrete example with a figure to illustrate the steps of the quantized k-means algorithm. The Appendix will contain additional background information on clustering to make the paper more self-contained. We will also integrate the four design principles of efficient deletion more tightly into the rest of the paper. In particular, we will discuss how the quantized k-means and divide-and-conquer k-means implement these principles. These edits will make the paper more accessible to the broad ML community. Thanks also for pointing out the typos; we will correct these.

**Response to Reviewer 3:**    Thank you for your very helpful suggestions. We focus on clustering because it's widely-used in applied ML, including on the UK Biobank dataset [1], and it's a good illustration of our deletion definition and design principles. We will add more real-world context and references for this point. For example, the cited work by Galinsky et al. makes uses of k-means clustering: "In this study, analyses of 113,851 UK Biobank samples showed that population structure in the UK is dominated by five principal components (PCs) spanning six clusters: Northern Ireland, Scotland, northern England, southern England, and two Welsh clusters."

You bring up an excellent point concerning the conceptual differences between deleting a single data point and deleting all the data pertaining to a single user. As you have pointed out, in production software stacks, such as at large internet companies, user data may exist in many databases, in many models, and in many relational formats (even between users and platforms). In production systems, it may indeed be the case that a deletion request from a single user's data may require the deletion of multiple points from multiple databases and multiple models in the system/datacenter. This is a promising direction for future research in data management for AI systems. We have clarified this distinction in our manuscript, as well as highlighted this as another topic for future work. Furthermore, even determining what pieces of data should get deleted in response to a user request is an interesting question.

Furthermore, thanks for your suggestion that we might compare the runtime of our algorithms to the standard k-means, which runs in time $O(nkTd)$. Both of our proposed methods run in fractional power time in $n$ (i.e. $\sqrt{n}$). We will state this comparison explicitly in our paper.

**Specific Response to Reviewer 4:**    Thank you for your constructive feedback. We will add a centralized section on related works in the main text; currently the discussion of relevant works are dispersed in a few places (e.g. in the discussion of the distinction from differential privacy). We agree that supervised learning is an important direction of future work for data deletion. We had some brief comments on efficient deletion for linear regression in the original submission. We will flesh this out as an illustration of how efficient deletion could work in supervised setting.

# References

[1] Kevin J Galinsky, Po-Ru Loh, Swapan Mallick, Nick J Patterson, and Alkes L Price. Population structure of uk biobank and ancient eurasians reveals adaptation at genes influencing blood pressure. *The American Journal of Human Genetics*, 99(5):1130–1139, 2016.


[Meta-Review · NeurIPS 2019]

The reviewers uniformly agreed this was a well-written, interesting, and novel paper. After reading the rebuttal, this general opinion did not change. Please take care to address all reviewers' (in particular R3's) post-rebuttal comments. Well done!